# AltLoRA: Towards Better Gradient Approximation in Low-Rank Adaptation with Alternating Projections

**Xin Yu** †
Department of Statistics
The Pennsylvania State University
State College, PA 16803
xmhy5152@psu.edu

**Yujia Wang**
College of Information Sciences
and Technology
The Pennsylvania State University
State College, PA 16803
yjw5427@psu.edu

**Jinghui Chen**
College of Information Sciences
and Technology
The Pennsylvania State University
State College, PA 16803
jzc5917@psu.edu

**Lingzhou Xue**‡
Department of Statistics
The Pennsylvania State University
State College, PA 16803
lzxue@psu.edu

## Abstract

Low-Rank Adaptation (LoRA) has emerged as an effective technique for reducing memory overhead in fine-tuning large language models. However, it often suffers from sub-optimal performance compared with full fine-tuning since the update is constrained in the low-rank space. Recent variants such as LoRA-Pro attempt to mitigate this by adjusting the gradients of the low-rank matrices to approximate the full gradient. However, LoRA-Pro's solution is not unique, and different solutions can lead to significantly varying performance in ablation studies. Besides, to incorporate momentum or adaptive optimization design, approaches like LoRA-Pro must first compute the equivalent gradient, causing a higher memory cost close to full fine-tuning. A key challenge remains in integrating momentum properly into the low-rank space with lower memory cost. In this work, we propose AltLoRA, an alternating projection method that avoids the difficulties in gradient approximation brought by the joint update design, meanwhile integrating momentum without higher memory complexity. Our theoretical analysis provides convergence guarantees and further shows that AltLoRA enables stable feature learning and robustness to transformation invariance. Extensive experiments across multiple tasks demonstrate that AltLoRA outperforms LoRA and its variants, narrowing the gap toward full fine-tuning while preserving superior memory efficiency.

## 1 Introduction

Low-Rank Adaptation (LoRA [25]) has emerged as a leading approach for parameter-efficient fine-tuning (PEFT)([24, 38, 35]) of large language models ([5, 51, 61, 40]). Building on prior work investigating the intrinsic dimensionality of neural networks ([2, 36]), LoRA assumes that fine-tuning updates can be effectively captured in a low-rank subspace. Specifically, for a pre-trained model with weight matrix $W_0 \in \mathbb{R}^{k \times d}$, LoRA reparameterizes the weight update $\Delta W$ via a low-rank

---

†First Author.

‡Correspondence to: Lingzhou Xue<lzxue@psu.edu>.

39th Conference on Neural Information Processing Systems (NeurIPS 2025).

decomposition as $W_0 + \Delta W = W_0 + sBA$, where $B \in \mathbb{R}^{k \times r}$, $A \in \mathbb{R}^{r \times d}$ and $s = \frac{\alpha}{r}$ is a scaling factor. Here, $r \ll \min(k, d)$ is the rank of the update. Thanks to its substantial memory and computational savings [25], LoRA has enabled scalable adaptation across diverse applications, including reinforcement learning from human feedback (RLHF) [57, 23], diffusion models [43, 77], and mixture-of-experts (MoE) architectures [67, 37].

Despite its parameter efficiency, LoRA often underperforms full fine-tuning ([13, 25, 41, 71]). This gap has fueled growing interest in optimizing LoRA via hyperparameter tuning under stable feature learning [21, 20] and optimizers that preserve transformation invariance [79]. Formally, if we denote the loss function as $L$, full fine-tuning will utilize the full gradient $\nabla_W L \in \mathbb{R}^{k \times d}$ for backpropagation. In contrast, the gradients in LoRA for $B$ and $A$ are given by $(\nabla_W L)A^\top$ and $B^\top(\nabla_W L)$, respectively (see Section 2). This reparameterization significantly alters the gradient flow during training [88] by restricting it to the low-rank space.

A promising direction to fill the gap between the gradient dynamics is to ensure that the equivalent gradient established by LoRA approximates the full gradient ([66, 65, 50]). However, two key challenges in the gradient approximation for low-rank adaptation remain unaddressed. First, LoRA-Pro [66] depends on an auxiliary variable that impacts the performance significantly. Depending on the choice of this variable, the evaluation score varies from $31.74$ to $57.57$ on the GSM8K datasets (see Appendix D.1 in [66]). Obtaining a unique solution requires solving a Sylvester equation, which introduces additional computational cost and relies on a non-standard assumption. Second, as LoRA-Pro accelerates the equivalent gradient with full-parameter learning, it requires a memory cost like full fine-tuning with space complexity $\mathcal{O}(kd)$ as shown in Table 1. In contrast, LoRA maintains a more efficient space complexity of $\mathcal{O}(kr + rd)$. Under such memory constraints, how to incorporate momentum properly within the low-rank structure is largely unexplored.

In this paper, to close the performance gap between LoRA and full fine-tuning, we address the two key challenges outlined above and propose a novel PEFT method, AltLoRA, based on **Alt**ernating updates to the **Lo**w-**R**ank **A**daptation. AltLoRA properly approximates the full gradient by alternately projecting it onto low-rank subspaces and $B$. Building on this projection-based gradient approximation, we further introduce a new mechanism to optimize momentum effectively within the low-rank space, while strictly adhering to the memory constraints of LoRA [25]. Without allowing full-parameter learning, AltLoRA is the first work in the literature to properly optimize both gradient and momentum over the low-rank subspaces, while achieving stable feature learning and transformation invariance, as summarized in Table 1.

Table 1: Comparison with Existing Work

| Methods | Gradient Approximation | Stable Feature Learning | Transformation Invariance | Time Complexity | Space Complexity |
|---|---|---|---|---|---|
| LoRA [25] | ✗ | ✗ | ✗ | $\mathcal{O}(kr^2 + dr^2)$ | $\mathcal{O}(kr + dr)$ |
| LoRA+ [21] | ✗ | ✓ | ✗ | $\mathcal{O}(kr^2 + dr^2)$ | $\mathcal{O}(kr + dr)$ |
| ScaledAdam [81] | ✗ | ✓ | ✗ | $\mathcal{O}(kr^2 + dr^2)$ | $\mathcal{O}(kr + dr)$ |
| LoRA-Rite [79] | ✗ | ✓ | ✓ | $\mathcal{O}(kr^2 + dr^2)$ | $\mathcal{O}(kr + dr)$ |
| LoRA-Pro [66] | ✓ | ✓ | ✓ | $\mathcal{O}(kdr)$ | $\mathcal{O}(kd)$ |
| AltLoRA | ✓ | ✓ | ✓ | $\mathcal{O}(kr^2 + dr^2)$ | $\mathcal{O}(kr + dr)$ |

Our main contributions are summarized as follows:

- We propose **AltLoRA**, a novel PEFT method that efficiently approximates the full gradient via alternating projections onto the low-rank subspaces $A$ and $B$. Moreover, we design a new momentum mechanism that operates within LoRA's memory constraints, enabling effective optimization of momentum within the low-rank space.

- Theoretically, we prove that AltLoRA ensures stable feature learning in the infinite-width neural network regime and, more generally, maintains transformation invariance, even when incorporating momentum. We also provide convergence guarantees for fine-tuning overparameterized two-layer ReLU networks.

- Empirically, we show the effectiveness of AltLoRA through extensive experiments on tasks including natural language understanding, dialogue generation, mathematical reasoning, and code generation. AltLoRA consistently outperforms existing LoRA-based methods.

## 2 Preliminary

Let us first revisit the optimization paradigm of LoRA [25]. If we denote the loss function as $L$, i.e., $L(A, B) := L(W + sBA)$, we can derive the gradient w.r.t $A$ and $B$ as follows:

$$\nabla_A L := \frac{\partial L}{\partial A} = \frac{\partial L}{\partial W}\frac{\partial W}{\partial A} = sB^T(\nabla_W L), \quad \nabla_B L := \frac{\partial L}{\partial B} = \frac{\partial L}{\partial W}\frac{\partial W}{\partial B} = s(\nabla_W L)A^T. \quad (1)$$

Here, as the full gradient is multiplied by the low-rank matrices to constitute the gradient of LoRA, it implicitly compresses the full gradient into the low-rank spaces. Suppose we use gradient descent to update $A$ and $B$, then the model parameter in the $(t+1)$-th iteration is:

$$\begin{aligned} W_{t+1} &= W_0 + sB_{t+1}A_{t+1} \\ &\approx W_0 + sB_t A_t - s\eta(\nabla_{B_t} L)A_t - s\eta B_t(\nabla_{A_t} L) \\ &= W_t - s\eta(\nabla_{B_t} L)A_t - s\eta B_t(\nabla_{A_t} L). \end{aligned} \quad (2)$$

Here, we omit the term related to $\eta^2$. Compared with the full gradient update $-\eta\nabla_W L$, LoRA's gradient can approximate the full gradient as long as $sB(\nabla_A L) + s(\nabla_B L)A$ is close to $\nabla_W L$. With a similar motivation, some previous work analyzes the approximation based on the Frobenius norm ([65, 66, 50]). Noticeably, LoRA-Pro [66] achieves gradient approxiation by adjusting the gradients of matrices $A$ and $B$ based on the following solutions:

$$g^A = \frac{1}{s}(B^T B)^{-1}B^T(\nabla_W L) + XA, g^B = \frac{1}{s}[I - B(B^T B)^{-1}B^T](\nabla_W L)A^T(AA^T)^{-1} - BX, \quad (3)$$

where $X \in \mathbb{R}^{r \times r}$ denotes an ancillary matrix and its selection is crucial and challenging for LoRA-Pro. As shown in their ablation studies, the selection of $X$ would vary the performance of the evaluation significantly. Besides, to obtain a unique solution for $X$, LoRA-Pro imposes additional uncommon assumptions to solve a Sylvester equation. However, even selecting a unique $X$, the equivalent gradient($sBg^A + sg^B A$) established by LoRA-Pro is independent of $X$, which implies that $X$ is only used to distinguish the gradient of $A$ and $B$ when jointly updating and doesn't influence the model update. It motivates the development of a more efficient alternating and eliminates the influence of $X$. To circumvent the ambiguity and inefficiency introduced by this joint updating strategy, we propose an alternating update strategy that approximates the full gradient as long as $sB(\nabla_A L)$ or $s(\nabla_B L)A$ is close to $\nabla_W L$.

**Notation.** Hereafter, we use the following notation to describe the asymptotic behavior as the width $n$ grows. Given sequences $c_n \in \mathbb{R}$ and $d_n \in \mathbb{R}^+$, we write $c_n = \mathcal{O}(d_n)$, resp. $c_n = \Omega(d_n)$, to refer to $c_n < \kappa d_n$, resp. $c_n > \kappa d_n$, for some constant $\kappa > 0$. For vector and matrix sequences, the notation is applied entry-wise. Additionally, we use $\odot$ and $\oslash$ to denote element-wise matrix multiplication and division, respectively. $[P]$ denotes the set of indices $\{1, \cdots, P\}$.

## 3 Methodology

### 3.1 Alternately Approximating the Full Gradient via Low-Rank Adaptation

We propose an alternating update scheme, where we update $A$ first and then update $B$ based on the new $A$. Define the low-rank modules as $A_t$ and $B_t$ at the $t$-th iteration, and the approximated gradients as $\tilde{\nabla}_A L$ and $\tilde{\nabla}_B L$, respectively. We begin by obtaining the optimal scaling gradient of $A$ by solving

$$\min_{\tilde{\nabla}_{A_t} L} \|sB_t(\tilde{\nabla}_{A_t} L) - \nabla_{W_t} L\|_F^2, \quad (4)$$

where $\| \cdot \|_F^2$ denotes the Frobenius norm squared—sum of squares of all entries in the matrix. Then by gradient descent, we can update $A$ and the full model as

$$A_{t+1} \leftarrow A_t - \eta\tilde{\nabla}_{A_t} L, \quad W_{t+\frac{1}{2}} \leftarrow W_t - \eta B_t(\tilde{\nabla}_{A_t} L), \quad (5)$$

where we update the full model at $(t + 1/2)$-th iteration to keep consistent with the joint update [66] (update $A$ and $B$ in one iteration). In our experiment, without any ambiguity, we treat the update $A$ or

$B$ as a single step (see Algorithm 1). After doing backpropagation w.r.t $A$, the gradient of $B$ doesn't approximate the full gradient at time $t$ since the full model has been update to the state of $(t + 1/2)$. Then we minimize the discrepancy between the full gradient at $W_{t+\frac{1}{2}}$ and the approximating gradient constructed by $B_t$ as follow

$$\min_{\tilde{\nabla}_{B_t} L} \| s(\tilde{\nabla}_{B_t} L) A_{t+1} - \nabla_{W_{t+\frac{1}{2}}} L \|_F^2. \tag{6}$$

Then by gradient descent, we can update $B$ and the full model as

$$B_{t+1} \leftarrow B_t - \eta \tilde{\nabla}_{B_t} L, \quad W_{t+1} \leftarrow W_{t+\frac{1}{2}} - \eta(\tilde{\nabla}_{B_t} L) A_{t+1}. \tag{7}$$

The following theorem gives the closed-form solution of Problems (4) and (6).

**Theorem 1.** *Assume $B_t \in \mathbb{R}^{k \times r}$ and $A_t \in \mathbb{R}^{r \times d}$ are full rank for any $t$, i.e. rank($B_t$) = rank($A_t$) = $r$. Solving Problems (4) and (6) yields the unique closed-form solutions*

$$
\begin{aligned}
\tilde{\nabla}_{A_t} L &= \frac{1}{s}(B_t^T B_t)^{-1} B_t^T (\nabla_{W_t} L) = \frac{1}{s^2}(B_t^T B_t)^{-1} \nabla_{A_t} L \\
\tilde{\nabla}_{B_t} L &= \frac{1}{s}\left(\nabla_{W_{t+\frac{1}{2}}} L\right) A_{t+1}^T (A_{t+1} A_{t+1}^T)^{-1} = \frac{1}{s^2} \nabla_{B_t} L (A_{t+1} A_{t+1}^T)^{-1},
\end{aligned}
\tag{8}
$$

*where $\nabla_{A_t} L$ and $\nabla_{B_t} L$ are the gradients of LoRA defined in Equation (1).*

Theorem 1 shows that both problems admit unique optimal solutions for $\tilde{\nabla}_{A_t} L$ and $\tilde{\nabla}_{B_t} L$, which only requires full rank. Therefore, it offers a new gradient approximation with less computational cost and promotes a more efficient updating strategy. Besides, instead of accessing the full gradient like full fine-tuning, the optimal gradient approximation only requires the standard gradient of $A$ or $B$ by backpropagation at each step and calculating the inverse of a small matrix with size $r \times r$.

Theorem 1 requires that the matrix $B_t$ and $A_t$ are full rank, but in the over-parameterized cases, the assumption is hard to achieve. To alleviate it, if we penalize the Frobenius norm of these two approximated gradients, i.e., weight decay, the condition can be eliminated (see Corollary 1). For simplicity, in the rest of the paper, we focus on the modified gradient in (8) for analysis. The closed-form solution in (8) yields the following full model update(with gradient descent)

$$
\begin{aligned}
W_{t+1} &= W_{t+\frac{1}{2}} - \eta(\tilde{\nabla}_{B_t} L) A_{t+1} \\
&= W_{t+\frac{1}{2}} - \eta(\nabla_{W_{t+\frac{1}{2}}} L) A_{t+1}^T (A_{t+1} A_{t+1}^T)^{-1} A_{t+1} \\
&= W_t - \eta B_t \tilde{\nabla}_{A_t} L - \eta(\nabla_{W_{t+\frac{1}{2}}} L) A_{t+1}^T (A_{t+1} A_{t+1}^T)^{-1} A_{t+1} \\
&= W_t - \eta B_t (B_t^T B_t)^{-1} B_t^T (\nabla_{W_t} L) - \eta(\nabla_{W_{t+\frac{1}{2}}} L) A_{t+1}^T (A_{t+1} A_{t+1}^T)^{-1} A_{t+1} \\
&= W_t - \eta Proj_{c(B_t)}(\nabla_{W_t} L) - \eta(\nabla_{W_{t+\frac{1}{2}}} L) Proj_{r(A_{t+1})}.
\end{aligned}
\tag{9}
$$

Interestingly, the proposed solution for gradient approximation in (8), is consistent with the literature work [59, 83, 73, 30, 42] called scaled gradient descent [46, 45] in low-rank matrix estimation [54]. Therefore, the view of gradient approximation would provide a novel interpretation of applying scaled gradient descent within the broader context of low-rank matrix decomposition. As optimizing LoRA with momentum for acceleration is a standard way in the literature [8, 25, 21], we will discuss how to properly design momentum within the low-rank space inspired by gradient approximation.

## 3.2 Proper Momentum Design within the Low-Rank Subspaces

For LoRA [25] and its variants [21, 86] without allowing full-parameter learning, the parameterization restricts both the gradient and the momentum updates to low-rank subspaces as the memory cost is $\mathcal{O}(kr + dr)$. As we have shown, the optimal gradient approximation under this constraint is obtained by projecting the full gradient onto the low-rank subspace. This insight naturally motivates the need to also align the momentum optimally within the same low-rank space, in order to fully leverage momentum-based acceleration under low-rank constraints.

Since the momentum evolves throughout training, it is essential to dynamically optimize it. For simplicity, we focus on the optimization paradigm for $B$ and develop our method inductively. Given

the aligned momentum $M_t^B$ within the low-rank space $A_t$ at time $t$, the alternating update strategy proceeds by updating $A$ to $A_{t+1}$ and then aligning $M_t^B$ with the new low-rank space $A_{t+1}$. To this end, we first recover $M_t^B$ to the full-dimensional space, and then project it onto the new subspace spanned by $A_{t+1}$, like gradient approximation. The following theorem formalizes this key idea.

**Theorem 2.** *Assume $A_{t+1}A_{t+1}^T$ is full-rank, i.e., $rank(A_{t+1}A_{t+1}^T) = r$. If $M_t^B$ has aligned with the low-rank space $A_t$ in the $t$-th iteration, by minimizing the following problem*

$$\min_{\tilde{M}_t^B} \|M_t^B A_t - \tilde{M}_t^B A_{t+1}\|_F^2. \tag{10}$$

*We can find $\tilde{M}_t^B = M_t^B A_t A_{t+1}^T (A_{t+1}A_{t+1}^T)^{-1}$, which makes the momentum aligned with the new low-rank space $A_{t+1}$ optimally.*

Theorem 2 shows that it is only necessary to store two small matrices so that we can optimize momentum properly. Similarly to Section 3.1, we can also remove the assumption of full rank here (see Corollary 2). In contrast to LoRA-Pro with full-parameter learning (Space Complexity $\mathcal{O}(kd)$), we aim to strictly satisfy the space complexity $\mathcal{O}(kr + dr)$ for parameter efficiency and keep momentum adaptively aligned with the low-rank spaces as the gradient approximation does.

A similar notion of momentum design is explored in [18, 22], where down-projection and up-projection matrices are employed to transfer compressed gradients across low-rank spaces. In contrast, we derive the optimal alignment directly within the low-rank subspaces to preserve gradient information. In Section 4.2, we theoretically demonstrate that aligning momentum with low-rank space guaranties formation invariance, whereas LoRA [25] and its variants [21, 86] have misaligned momentum that undermines this robustness [79].

After analyzing how to efficiently optimize both the gradient and momentum under limited resource constraints, we summarize our proposed algorithm, AltLoRA, in Algorithm 1. Unlike the joint update strategy, AltLoRA updates only one of the low-rank matrices, either $A$ or $B$, at each step, based on the scaled gradient and momentum presented in Theorems 1 and 2. The number of trainable parameters at each step is reduced by half compared to the joint update. Designed as a practical PEFT method, AltLoRA can be seamlessly integrated into existing libraries such as Hugging Face [69] (see Appendix C.1 for implementation details). To further accelerate and stabilize the training paradigm of AltLoRA, we introduce AltLoRA+, an enhanced variant that naturally incorporates second-moment estimates similar to AdamW (see Algorithm 2 for details).

---

**Algorithm 1:** AltLoRA: Gradient Approximation via Alternating Projection with Proper Momentum Design under LoRA's Memory Constraint

---

**Input:** Momentum states $M_0^A$, $M_0^B$; scaling factor $s = \frac{\alpha}{r}$; learning rate $\eta$; momentum coefficient $\beta_1$; total steps $T$; weight decay $\gamma$
**Output:** Final matrices $A_T$ and $B_T$

**for** $t = 0, \ldots, T-1$ **do**
    **if** $t \bmod 2 = 0$ **then**
        **Update $A$:**
            Only backpropagate w.r.t. $A_t$ and obtain $\nabla_{A_t} L$
            $\tilde{\nabla}_{A_t} L = \frac{1}{s^2}(B_t^\top B_t)^{-1} \nabla_{A_t} L$
            $\tilde{M}_t^A = (B_t^\top B_t)^{-1} B_t^T B_{t-1} M_{t-1}^A$
            $M_t^A \leftarrow \beta_1 \tilde{M}_t^A + (1-\beta_1)\tilde{\nabla}_{A_t} L$
            $A_{t+1} \leftarrow A_t - \eta(M_t^A + \gamma A_t)$
    **else**
        **Update $B$:**
            Only backpropagate w.r.t. $B_t$ and obtain $\nabla_{B_t} L$
            $\tilde{\nabla}_{B_t} L = \frac{1}{s^2} \nabla_{B_t} L (A_{t+1}A_{t+1}^\top)^{-1}$
            $\tilde{M}_t^B = M_{t-1}^B A_t A_{t+1}^\top (A_{t+1}A_{t+1}^\top)^{-1}$
            $M_t^B \leftarrow \beta_1 \tilde{M}_t^B + (1-\beta_1)\tilde{\nabla}_{B_t} L$
            $B_{t+1} \leftarrow B_t - \eta(M_t^B + \gamma B_t)$

---

**Time Complexity and Space Complexity.** When $r \ll min\{k, d\}$, the time and memory cost of AltLoRA and AltLoRA+ is similar to the standard LoRA and more efficient compared with LoRA-

Pro. The additional computational cost takes $\mathcal{O}(r^3)$ time, and since $r$ is very small, this overhead is negligible when compared with the back-propagating time. In the experiment, we will show that the delay time compared with LoRA is mild even when the rank $r$ increases. (see Table 3).

## 4 Theoretical Analysis

### 4.1 Stable Feature Learning

Given the current trend of increasing model sizes ([76, 47, 75]), it raises a lot of attention to analyze the asymptotic training behavior of neural networks as the number of neurons approaches infinity ([56, 19, 74]). There is a line of work in LoRA ([21, 20, 81]) considering the infinite-width NN setting. To achieve stable feature learning (see Definition 2 in Appendix D.1), they propose a fine-grained choice of hyperparameters in the original LoRA, like the learning rate [21], the initialization ([20]), and the optimizer ([81]). The core idea is that the update increment over the loss function or parameter should be of constant magnitude, which ensures that neither the NN predictions nor the increments explode or vanish as the NN size increases, thereby leading to stable training dynamics. First, we demonstrate that our method achieves stable feature learning on a toy model in Appendix D.1.1. We then prove that this stability extends to arbitrary LoRA ranks and holds for AltLoRA and AltLoRA+, which we formalize in the theorem below. For clarity of presentation, we omit the scaling factor $s$ in the subsequent theorems and analysis.

**Theorem 3** (Informal). *Assume that, with the input $x$, $BAx$ has dimension $\mathcal{O}(n)$. In Algorithm 1 or Algorithm 2, if we use the same learning rate $\eta = \mathcal{O}(1)$ to update $A$ and $B$, it would achieves stable feature learning. Moreover, without momentum in AltLoRA or AltLoRA+, the model update achieves stable feature learning as well with*

$$W_{t+1} = W_t - \eta Proj_{c(B_t)}(\nabla_{W_t} L) - \eta(\nabla_{W_{t+\frac{1}{2}}} L)Proj_{r(A_{t+1})}, \tag{11}$$

*where*

$$\eta Proj_{c(B_t)}(\nabla_{W_t} L), \eta(\nabla_{W_{t+\frac{1}{2}}} L)Proj_{r(A_{t+1})} \in \mathcal{O}(1).$$

*However, when doing joint update ([81]), the update will introduce additional across term $\eta^2(\nabla_{W_t} L)A_t^T(A_t A_t^T)^{-1}(B_t^T B_t)^{-1}B_t^T(\nabla_{W_t} L) \in \mathcal{O}(1)$. The across term is indeed the second order term w.r.t $\eta$, but it is same magnitude as $\eta(\nabla_{W_t} L)Proj_{r(A_t)}$ and $\eta Proj_{c(B_t)}(\nabla_{W_t} L)$ in infinite-width NN setting.*

In Theorem 3, AltLoRA and AltLoRA+ achieve stable feature learning. Moreover, as the joint update would introduce the cross term with an unignorable magnitude (especially $\eta$ is $\mathcal{O}(1)$ instead of $\mathcal{O}(1/n)$ in the toy model), joint update with scaled gradient descent ([81]) breaks the clean interpretation of projecting the full gradient onto low-rank subspaces and degrade the performance as our experiment studies show later.

### 4.2 Transformation Invariance

With the motivation that an optimizer should yield the same update to the full model regardless of the specific factorization, transformation invariance, as a sufficient condition for stable feature learning, is proposed by LoRA-RITE [79]. Here, we will prove that our designed gradient and momentum in Algorithm 1 would be inherently robust as transformation invariance.

**Definition 1.** *If there are two pairs of LoRA matrix $(A_1, B_1), (A_2, B_2)$ can represent the same finetuned weight $W = W_0 + B_1 A_1 = W_0 + B_2 A_2$. An optimizer exhibits transformation invariance if its updates, $(\delta A_1, \delta B_1)$ and $(\delta A_2, \delta B_2)$ satisfy*

$$\begin{align} W_0 + (B_1 + \delta B_1)(A_1 + \delta A_1) = W_0 + (B_2 + \delta B_2)(A_2 + \delta A_2) \\ \Rightarrow (B_1 + \delta B_1)(A_1 + \delta A_1) = (B_2 + \delta B_2)(A_2 + \delta A_2). \end{align} \tag{12}$$

LoRA-RITE [79] notices that, after combining scaled gradient descent with element-wise Adam in [81], the ScaledAdam can't preserve transformation invariance. As the momentum is optimized properly, we will analyze how AltLoRA keeps transformation invariance naturally, especially when incorporating momentum.

Recall the definition of projection matrices in Equation (9): $Proj_{c(B_t)} := B_t(B_t^T B_t)^{-1} B_t^T$ (or $Proj_{r(A_t)} := A_t^T (A_t A_t^T)^{-1} A_t$). The following lemma provides insight into how Algorithm 1 achieves transformation invariance.

**Lemma 1.** *If any two pairs of LoRA factors* $(A_1, B_1), (A_2, B_2)$ *satisfying*

$$W = W_0 + B_1 A_1 = W_0 + B_2 A_2, \tag{13}$$

*then* $$Proj_{c(B_1)} = Proj_{c(B_2)}, \quad Proj_{r(A_1)} = Proj_{r(A_2)}.$$

Even though the full model update can be decomposed into different pairs of low-rank adaptations, within each pair of LoRA factors, the column space of $B$ (or the row space of $A$) is equivalent to the column space (or the row space) of the full model update. Therefore, the projection matrix would be preserved invariant over the pairs of low-rank adaptation.

**Theorem 4.** *AltLoRA in Algorithm 1 is transformation-invariant.*

Building on the insight from Lemma 1, we leverage the invariance of the projection matrix to the low-rank subspaces to approximate the full gradient via the gradient and moment information. As a result, with the goal of gradient approximation without full-parameter learning, our method achieves transformation invariance inherently. LoRA-RITE [79] is also aware of the equivalence of low-rank spaces, but they do not notice or exploit the invariance of the projection matrix. Instead, they design an unmagnified gradient requiring polar decomposition at each iteration, which introduces additional computational overhead. In contrast, our method avoids polar decomposition, contributing to its superior efficiency (see Table 3). LoRA-Pro [66] also achieves transformation invariance but does so without adhering to LoRA's memory constraint. AltLoRA in Algorithm 1, by comparison, strictly follows the memory budget of LoRA while preserving transformation invariance through a more efficient design. While Algorithm 2 does not currently maintain transformation invariance under second-order momentum, this opens an exciting avenue for future research. In Appendix D.2, we provide a detailed discussion on why extending our first-order momentum design to the second order poses fundamental challenges. Despite this, AltLoRA+ achieves substantial empirical gains over LoRA and its variants, demonstrating the practical strength of our approach even when we only keep the transformation-invariant up to the second momentum.

### 4.3 Convergence Analysis

Following [81], we provide a convergence analysis of AltLoRA (or AltLoRA+) without momentum within the over-parameterized two-layer ReLU NN tuning problem (see Appendix D.3). In Theorem 7, we show that the convergence is independent of the condition number of the data matrix. In contrast to [81], we impose fewer assumptions to establish the convergence analysis. Notably, we don't require the extended spectral initialization in Definition 7.3 [81]. In our experimental study, AltLoRA (AltLoRA+) can achieve superior performance with the variant of initialization used by LoRA and its variants (see Appendix E.3.2), which supports our insight empirically.

## 5 Experimental Results

This section empirically shows the effectiveness of our approach across various model architectures and datasets. Section 5.1 summarizes the experimental settings and results on supervised fine-tuning (SFT) benchmark tasks, and Section 5.2 provides details of the setup and results for natural language understanding tasks. Finally, ablation studies from multiple perspectives are presented in Section 5.3. The code for our project is available at https://github.com/LucasXinYu/AltLoRA.

### 5.1 Experiments on SFT of LLM: Natural Language Generation

**Training Details.** We assess our methods on dialogue generation with the WizardLM dataset [72], mathematical reasoning with the MetaMathQA dataset [80], and code generation with the CodeFeedBack dataset [90] using the LLama-3.1-8B and Llama-3-8B models [17] (see Appedix E.1). We compare AltLoRA and AltLoRA+ with the pretrained model, full fine-tuning, LoRA [25], PisSSA[44], rsLoRA[31], LoRA+[21], DoRA[41], AdaLoRA[86], LoRA-GA[65], LoRA-Rite [79]and LoRA-Pro[66]. To ensure fair comparisons, we closely follow the experimental protocol established by [66]. Unless otherwise stated, we fine-tune models using default hyperparameters (if

used): $\beta_1 = 0.9$, $\beta_2 = 0.999$, and zero weight decay. We adopt a cosine learning rate schedule with a warm-up ratio of 0.03. LoRA adapters are applied to $\{Q, K, V, O\}$ layers. By default, we set the rank to $r = 8$ and the scaling factor to $\alpha = 32$ for dialogue generation tasks, and $r = 8$, $\alpha = 16$ for the mathematical reasoning and code generation tasks. We carefully grid search the learning rates [‡]. To obtain a reliable estimate of model performance, we perform three runs with different random seeds and report the average and standard deviation of the results.

**Evaluations.** We evaluate the baselines similar to [66]. Specifically, for the dialogue generation task, we use the MT-Bench dataset [89] with GPT-4o, with scores ranging from 1 to 10. We report the score from the first turn as our metric. For the math task, we evaluate the model on the GSM8K test set [11] using the LLM Evaluation Harness [16], and we report the exact match accuracy. For the code generation task, we evaluate on the HumanEval dataset [6] and report the PASS@1 metric.

**Results.** Table 2 presents our experimental results, which demonstrates AltLoRA superior performance. With a rank of 8, AltLoRA achieves noticeable improvement over the original LoRA: 0.5 on MT-bench, 8.38 on GSM8K and 3.1 on HumanEval using Llama-3.1-8B. Notably, AltLoRA achieves significantly higher scores on MT-Bench compared to LoRA-Pro and Full FT. In addition, AltLoRA+ yields improvements over LoRA-Pro on both GSM8K and HumanEval, and AltLoRA+ obtains better performance in mathematical reasoning than Full FT. These further demonstrate the effectiveness of the new design gradient and momentum. The additional study on Llama-3-8B model (see Table 5 in Appendix E.1) also demonstrates a clear advantage over baseline methods.

Table 2: Comparison of different LoRA variants on MT-Bench, GSM8K, and HumanEval benchmarks on Llama-3.1-8B-Base. **Bold** indicates the best result, underline represents the second-best one.

| Method | MT-Bench | GSM8K | HumanEval |
|---|---|---|---|
| PreTrain | 5.93±0.08 | 51.34±1.38 | 36.15±1.97 |
| Full FT | 6.31±0.04 | 73.31±0.32 | **50.81±1.10** |
| LoRA | 6.06±0.02 | 66.11±1.43 | 40.31±1.34 |
| PiSSA | 5.15±0.10 | 67.78±1.11 | 42.44±1.11 |
| rsLoRA | 6.10±0.06 | 68.12±0.44 | 43.91±1.44 |
| LoRA+ | 6.40±0.06 | 72.33±1.33 | 44.10±1.38 |
| DoRA | 6.08±0.03 | 68.33±0.88 | 42.13±1.31 |
| AdaLoRA | 6.08±0.05 | 72.63±1.45 | 42.21±2.66 |
| LoRA-GA | 6.00±0.09 | 70.33±0.91 | 42.01±1.21 |
| LoRA-Pro | 6.19±0.03 | 73.12±0.56 | 43.13±1.45 |
| LoRA-Rite | 6.10±0.01 | 74.10±0.31 | 43.12±0.51 |
| AltLoRA | **6.56±0.04** | 74.49±0.57 | 45.91±1.14 |
| AltLoRA (rank=32) | 6.39±0.04 | 73.24±0.29 | 46.87±1.49 |
| AltLoRA (rank=128) | 6.27±0.01 | 74.11±0.21 | 45.41±1.65 |
| AltLoRA+ | 6.16±0.02 | **76.91±0.31** | 50.10±1.35 |
| AltLoRA+ (rank=32) | 6.10 ±0.02 | 76.32±0.29 | 49.97±1.52 |
| AltLoRA+ (rank=128) | 6.07±0.03 | 77.08±0.83 | 49.77±1.58 |

**Memory and Time Consumptions.** In Table 3, we also compare the memory cost and training time of our methods with Full FT, LoRA, LoRA-Rite and LoRA-Pro on Llama-3.1-8b mode. Without full-parameter learning, we have a comparable memory cost and training time close to LoRA. After taking a higher rank of LoRA, the memory cost and computation cost won't increase significantly. However, as LoRA-Pro requires storing the full size first-order momentum and second-order momentum, it leads to an unignorable cost like Full FT. As LoRA-Rite incurs additional calculations like polar decomposition, it also increase the computation time.

---

[‡]See Appendix E.1 for details of learning rate grid search. We set the sequence length to 1024 and the macro batch size to 4 for math and code tasks, and macro batch size to 8 for dialogue generation. All experiments are conducted on NVIDIA A100 and NVIDIA A6000 GPUs.

Table 3: Comparison of memory usage and training time across different fine-tuning methods.

| Method | Memory Cost | Training Time |
|---|---|---|
| Full FT | > 48 GB | 4h 23min |
| LoRA | 22.26 GB | 2h 13min |
| LoRA-Rite | 25.39 GB | 2h 44min |
| LoRA-Pro | 40.12 GB | 4h 5min |
| AltLoRA | 22.56 GB | 2h 34min |
| AltLoRA(rank=32) | 23.11 GB | 2h 41min |
| AltLoRA(rank=128) | 25.11 GB | 2h 52min |
| AltLoRA+ | 23.16 GB | 2h 38min |
| AltLoRA+(rank=32) | 24.98 GB | 2h 45min |
| AltLoRA+(rank=128) | 27.76 GB | 2h 56min |

## 5.2 Experiments on Natural Language Understanding

**Training and Evaluation Details.** We assess our methods natural language understanding on a subset of GLUE benchmark dataset with fine-tuning a T5-base[52] model. We compare AltLoRA and AltLoRA+ with the full fine-tuning, LoRA [25], PisSSA[44], rsLoRA[31], LoRA+[21], DoRA[41], AdaLoRA[86], LoRA-GA[65], and LoRA-Pro[66]. We fine-tune the T5-based model [52] with our methods and the baselines on a subset of GLUE datasets [63]: MNLI, SST2, CoLA, QNLI, and MRPC. We use the accuracy as the evaluation metric. To ensure fair comparison, all experiments are run three times with different random seeds, and we report the mean and standard deviation of the results. Due to space constraints, additional experimental details are provided in Appendix E.1.

**Results.** As shown in Table 4, AltLoRA+ outperforms the baselines on average. In particular, it achieves the highest score on MRPC, the second-highest on CoLA, MNLI, and SST-2 datasets.

Table 4: Performance of fine-tuning T5-Base on 5 sub-tasks of the GLUE benchmark. **Bold** indicates the best result, underline represents the second-best one, and * marks results reported from [65].

| Method | MNLI | SST-2 | CoLA | QNLI | MRPC | Average |
|---|---|---|---|---|---|---|
| Full | **86.29**±**0.01** | 93.97±0.06 | 80.87±0.05 | 93.02±0.03 | 86.89±0.13 | 88.21 |
| LoRA | 85.32±0.01 | 93.76±0.05 | 81.31±0.20 | 92.96±0.09 | 86.03±0.24 | 87.88 |
| RSLoRA | 85.23±0.01 | 93.96±0.06 | 81.21±0.14 | 93.12±0.09 | 86.27±0.24 | 87.96 |
| DoRA | 85.58±0.03 | 93.65±0.06 | 81.16±0.04 | 93.04±0.06 | 86.14±0.12 | 87.91 |
| LoRA+ | 85.32±0.06 | 93.92±0.11 | 81.21±0.06 | 92.97±0.03 | 86.25±0.16 | 87.93 |
| PiSSA | 85.87±0.04 | 93.84±0.06 | **81.90**±**0.05** | 93.16±0.09 | 86.64±0.12 | 88.28 |
| LoRA-GA* | 85.70±0.09 | **94.11**±**0.18** | 80.57±0.20 | **93.18**±**0.06** | 85.29±0.24 | 87.77 |
| AdaLoRA | 85.45±0.11 | 93.92±0.09 | 80.31±0.05 | 91.66±0.05 | 86.16±0.60 | 87.50 |
| LoRA-Pro | 85.70±0.11 | 93.92±0.10 | 78.42±0.03 | 93.15±0.03 | 86.54±0.50 | 87.55 |
| AltLoRA | 85.26±0.04 | 93.87±0.05 | 80.44±0.09 | 91.56±0.01 | 86.60±0.99 | 87.55 |
| AltLoRA+ | 85.81±0.03 | 94.03±0.12 | 81.44±0.30 | 92.99±0.03 | **87.25**±**1.12** | **88.30** |

## 5.3 Ablation Study

Figure 1 presents an ablation study of the learning rate $\eta$ and the scaling factor $\alpha$ for LoRA, AltLoRA and AltLoRA+, using the LLaMA 3.1-8B model on mathematical reasoning tasks. The results show that our proposed methods are robust in learning rate and the scaling factor with consistent superior performance. Moreover, it shows that $\alpha = 16$ obtains overall better performance compared to $\alpha = 8$ and $\alpha = 32$. The influence of increasing rank is reported in Table 2 (see Appendix E.3 of the results on Llama-3-8B model). Besides, studying the choice of hyperparameters, in Appendix E.3.2, we present additional ablation studies on the Llama 3.1-8B model as well. To evaluate the effectiveness of alternating strategies, we compare them against the joint update method. As the approaches of multiple LoRA modules, such as in the mixture of LoRA experts, has gained popularity [37, 70], we also assess the impact of varying the number of experts in LoRA layers. Finally, to further validate the robustness of our method with respect to initialization, as discussed in Section 4.3, we study



Figure 1: Evaluation Accuracy of LoRA, AltLoRA and AltLoRA+ for various learning rate $\eta$ and scaling factor $\alpha$ combination on the GSM8K datasets using Llama-3.1-8B.

different initialization strategies. These ablation studies collectively demonstrate that our method is robust to hyperparameter variations and is applicable to more complex model architectures.

# 6 Conclusion

We propose AltLoRA, a memory-efficient fine-tuning method that alternates updates of low-rank matrices to dynamically project both the gradient and momentum within low-rank subspaces. By leveraging an efficient closed-form gradient approximation and a principled momentum design, AltLoRA operates entirely under low-rank constraints while ensuring stable feature learning and transformation invariance without requiring full-parameter learning. Extensive experiments across diverse tasks demonstrate the superior performance of AltLoRA and its enhanced variant, AltLoRA+, over LoRA and its variants, narrowing the gap to full fine-tuning while retaining memory efficiency.

## Acknowledgements

The work of X. Yu and L. Xue was supported by the U.S. National Science Foundation under the grants CCF-2007823 and DMS-2210775, and by the U.S. National Institutes of Health under the grant 1R01GM152812. The work of Y. Wang and J. Chen was partially supported by the National Science Foundation under Grant No. 2348541. The views and conclusions contained in this paper are those of the authors and should not be interpreted as representing any funding agencies.

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

# A  Related Work

Low-rank adaptation(LoRA)([25]) has been the subject of extensive research in foundation models([51, 5, 1, 33, 55, 61]), with numerous variations and improvements ([34, 28, 32, 78, 12, 27, 91]). One line of research focuses on dynamically adjusting the LoRA rank during training. This includes DyLoRA[62], IncreLoRA[82], and AdaLoRA[86]. Another line of work involves enhancing LoRA performance through the addition of extra scaling matrices, which include DoRA[41] and DeepLoRA[78]. These directions are orthogonal to our work. Regarding the optimization of LoRA, we find that the following topics are close to our work.

**Stable Feature Learning**  Under the infinite-width NN setting([19, 56]), LoRA+([21]) finds that the standard LoRA is inefficient and they propose to use different learning rates for $A$ and $B$. To provide a careful choice of hyperparameters for efficient use of LoRA, a line of work analyzes LoRA under efficient learning ([20, 81]). Noticeably, [81] introduces preconditioners under a Riemannian metric ([45]) and updates LoRA by using scaled gradients of $A$ and $B$ simultaneously. While their method aims to improve stability and efficiency, it is important to note that their goal is not to approximate the full gradient. This approach does not yield an optimal approximation to the full gradient update. Moreover, [79] proposes an adaptive matrix preconditioning method preserving transformation-invariant, a sufficient condition for stable feature learning.

**Approximation full-tuning or full gradient**  To fill the gap between LoRA and full fine-tuning, there are two lines of work with different motivations. The first class of work focuses on the initialization, like [66]. It proposes to make the initialization of LoRA align with the full-finetuning directly. However, after the first step, how difference between LoRA and full-tuning is unknown. The second line of work focuses on optimizing LoRA properly over the optimization trajectory([66, 50, 87]). Noticeably, [66] proposes to optimize the gradients of $A$ and $B$ together to approximate the full gradient. But the optimal approximation is hard to find under practical conditions and aligning momentum towards the full gradient requires storing a full-size matrix ($k \times d$) in their algorithm. These challenges also exist in later work ([50]).

**Gradient Projection in LoRA**  Motivated by the view that LoRA updates can be viewed as performing random projection from the full gradient, F-LoRA([18]) achieves high-rank updates by resampling the projection matrices. There are also some approaches that propose training networks with low-rank factorized weights from scratch ([64, 32]). Random projection is also applied in Ga-LoRA([88]) and following work([39, 9]), but they need to access the full model and can't store the low-rank adapter in the end. On the contrary, without full-parameter learning, we use gradient projection to keep the gradient best preserved in the low-rank spaces.

**Alternating Update**  To the best of our knowledge, we haven't found the existing work of updating LoRA alternately in the centralized setting, but in the decentralized setting, i.e., Federated Learning, we notice [7] used the alternating strategy to address the challenge of inaccurate model aggregation([68, 3, 58]) with computational and communication efficiency. Besides, in the centralized setting, [85] proposes to freeze $A$ and update $B$, which would be regarded as a specific case of our work to do alternating minimization.

**Scaled Gradient Descent**  Our proposed methods are also closely related to scaled gradient descent(Scaled GD) in traditional low-rank matrix estimation under over-parameterization and ill-conditioning ([59, 60, 29, 46]). Notably, [59] shows that the scaled GD would keep the convergence independent of the condition number. Different variants of scaled GD have been proposed and studied in work ([73, 83, 10, 84]). For the alternating scaled GD, [30] finds that it would enable faster convergence with larger step sizes compared with scaled GD. And [42] provably shows that alternating scaled GD would achieve a linear convergence rate, starting from arbitrary random initialization.

# B  The Proof and Details in Section 3

In this section, we provide the formal proofs and detailed discussions supporting the results presented in Section 3. Specifically, Appendix B.1 presents the proof of Theorem 1, removes the full-rank assumption in Corollary 1 via weight decay. Appendix B.2 contains the proof of Theorem 2 and demonstrates how the full-rank assumption can similarly be relaxed using weight decay in Corollary 2.

## B.1 The Proof in Section 3.1

### B.1.1 The Proof of Theorem 1

*Proof.* The first-order condition of Problem (4) yields

$$sB_t^T(sB_t\tilde{\nabla}_{A_t}L - \nabla_{W_t}L) = 0, \tag{14}$$

where $s$ is a positive scaling factor. Then we can reorganize it and obtain

$$sB_t^T B_t\tilde{\nabla}_{A_t}L = B_t^T\nabla_{W_t}L. \tag{15}$$

As we assume the matrix $B$ is full rank, it yields

$$\tilde{\nabla}_{A_t}L = \frac{1}{s}(B_t^T B_t)^{-1}B_t^T(\nabla_{W_t}L). \tag{16}$$

Furthermore, recalling the definition of the gradient of standard LoRA in (1), we obtain

$$\tilde{\nabla}_{A_t}L = \frac{1}{s}(B_t^T B_t)^{-1}B_t^T(\nabla_{W_t}L) = \frac{1}{s^2}(B_t^T B_t)^{-1}\nabla_{A_t}L. \tag{17}$$

Similarly, we can obtain the closed-form solution of $\tilde{\nabla}_{B_t}L$ in (8). $\qquad\square$

### B.1.2 Corollary 1 and Its Proof

**Corollary 1.** *For $B \in \mathbb{R}^{k\times r}$ and $A \in \mathbb{R}^{r\times d}$, solving problems in (18)*

$$\min_{\tilde{\nabla}_{A_t}L} \|sB_t(\tilde{\nabla}_{A_t}L) - \nabla_{W_t}L\|_F^2 + \frac{\lambda}{2}\|s\tilde{\nabla}_{A_t}L\|_F^2$$

$$\min_{\tilde{\nabla}_{B_t}L} \|s(\tilde{\nabla}_{B_t}L)A_{t+1} - \nabla_{W_{t+\frac{1}{2}}}L\|_F^2 + \frac{\lambda}{2}\|s\tilde{\nabla}_{B_t}L\|_F^2, \tag{18}$$

*yields the unique closed-form solution*

$$\tilde{\nabla}_{A_t}L = \frac{1}{s}(B_t^T B_t + \lambda\mathbb{I}_{r\times r})^{-1}B_t^T(\nabla_{W_t}L) = \frac{1}{s^2}(B_t^T B_t + \lambda\mathbb{I}_{r\times r})^{-1}\nabla_{A_t}L,$$

$$\tilde{\nabla}_{B_t}L = \frac{1}{s}(\nabla_{W_{t+\frac{1}{2}}}L)A_{t+1}^T(A_{t+1}A_{t+1}^T + \lambda\mathbb{I}_{r\times r})^{-1} = \frac{1}{s^2}\nabla_{B_t}L(A_{t+1}A_{t+1}^T + \lambda\mathbb{I}_{r\times r})^{-1}. \tag{19}$$

*where $\mathbb{I}_{r\times r}$ is the $r \times r$ identity matrix and $\lambda > 0$.*

*Proof.* For the first line problem in (18), the first-order condition yields

$$sB_t^T(sB_t\tilde{\nabla}_{A_t}L - \nabla_{W_t}L) + \lambda s^2\tilde{\nabla}_{A_t}L = 0, \tag{20}$$

where $s$ is a positive scaling factor. Then we can reorganize it and obtain

$$s(B_t^T B_t + \lambda\mathbb{I})\tilde{\nabla}_{A_t}L = B_t^T\nabla_{W_t}L. \tag{21}$$

To keep $(B_t^T B_t + \lambda\mathbb{I})$ invertible, we only require that $\lambda$ isn't too small and it yields

$$\tilde{\nabla}_{A_t}L = \frac{1}{s}(B_t^T B_t + \lambda\mathbb{I})^{-1}B_t^T(\nabla_{W_t}L). \tag{22}$$

Furthermore, recalling the definition of the gradient of standard LoRA in (1), we obtain

$$\tilde{\nabla}_{A_t}L = \frac{1}{s}(B_t^T B_t + \lambda\mathbb{I})^{-1}B_t^T(\nabla_{W_t}L) = \frac{1}{s^2}(B_t^T B_t + \lambda\mathbb{I})^{-1}\nabla_{A_t}L. \tag{23}$$

Similarly, we can obtain the closed-form solution of $\tilde{\nabla}_{B_t}L$ in (19). Noticeably, the result $(\nabla_{W_{t+\frac{1}{2}}}L)A_{t+1}^T = \nabla_{B_t}L$ holds with the fact that $W_{t+\frac{1}{2}} = W_0 + B_tA_{t+1}$. $\qquad\square$

In Corollary 1, the hyperparameter $\lambda$ can be small enough ($1e^{-6}$ in our numerical studies) and we don't tune the hyperparameter overall. For more discussion about the selection of $\lambda$ in the over-parameterized setting for low-rank matrix estimation, please refer to APGD([42]), ScaledGD([73]), and NoisyPrecGD([84]).

### B.2 Proof of Section 3.2

#### B.2.1 Proof of Theorem 2

*Proof.* The proof is similar to Theorem 1 thus we omit it here. $\qquad\square$

#### B.2.2 Corollary 2 and Its Proof

**Corollary 2.** *If we assume $M_t^B$ has aligned with the full gradient in the t-th iteration, by minimizing the following problem*

$$\min_{\tilde{M}_t^B} \|M_t^B A_t - \tilde{M}_t^B A_{t+1}\|_F^2 + \frac{\lambda}{2}\|\tilde{M}_t^A\|_F^2, \tag{24}$$

*we can find the unique solution $\tilde{M}_t^B = M_t^B A_t A_{t+1}^T (A_{t+1}A_{t+1}^T + \lambda\mathbb{I})^{-1}$, which is the best approximation of current full gradient.*

*Proof.* The proof is similar to Corollary 1 thus we omit it here. $\qquad\square$

## C Appendix for Algorithm 1

### C.1 The Implementing Details for Algorithm 1

AltLoRA, as a novel PEFT method, can be seamlessly integrated into popular libraries such as Hugging Face Transformers [69]. The key engineering modifications are as follows:

- **Alternating Updates**: To enable alternating optimization of LoRA parameters, we extend the existing Transformer architecture by introducing a control argument within the `training_step` function. This argument identifies the current update phase and selectively disables gradient computation for parameters named `"lora_A"` or `"lora_B"`, thereby facilitating an efficient alternating update mechanism.

- **Custom Optimizer Integration**: Similar to prior LoRA variants that incorporate new optimizers [81, 66], AltLoRA can be easily adapted by implementing a new optimizer class. This allows flexible modification of the optimization dynamics tailored to the alternating update strategy. It would provide a broader impact to incorporate with other parameter-efficient structures, like MoE or RLHF, when using low-rank adaptation.

### C.2 AltLoRA+

With the goal of approximating the full gradient under the memory constraint of standard LoRA, we propose AltLoRA in Algorithm 1 to properly optimize the training paradigm of LoRA. Furthermore, the ultimate goal is to fill the gap of performance between the existing parameter-efficient fine-tuning methods, like LoRA([25]), and the full model fine-tuning. Therefore, witnessing the success of incorporating the second momentum for accelerating and stabilizing the optimizing paradigm [25], we propose a variant of AltLoRA, called AltLoRA+ (see Algorithm 2) to help accelerate our optimizer with second momentum. The increasing memory cost for storing second momentum is $\mathcal{O}(kr + dr)$, so AltLoRA+ won't require storing the full size matrix $\mathcal{O}(kd)$ like LoRA-Pro [66].

---

**Algorithm 2:** AltLoRA+: AltLoRA with Second Order Momentum

---

**Input:** Momentum states $M_0^A$, $M_0^B$, $V_0^A$ and $V_0^B$, scaling factor $s = \frac{\alpha}{r}$, learning rate $\eta$,
momentum coefficient $\beta_1$ and $\beta_2$, total number of steps $T$, weight decay coefficient $\gamma$,
and constant $\epsilon$

**Output:** Final matrices $A_T$ and $B_T$

**for** $t = 0, \ldots, T-1$ **do**
    **if** $t \bmod 2 = 0$ **then**
        **Update** $A$**:**
            Only backpropagate w.r.t. $A_t$ and obtain $\nabla_{A_t} L$
            $\tilde{\nabla}_{A_t} L = \frac{1}{s^2} (B_t^\top B_t)^{-1} \nabla_{A_t} L$
            $\tilde{M}_t^A = (B_t^\top B_t)^{-1} B_t^\top B_{t-1} M_{t-1}^A$
            $M_t^A \leftarrow \beta_1 \tilde{M}_t^A + (1 - \beta_1) \tilde{\nabla}_{A_t} L$
            $V_t^A \leftarrow \beta_2 V_{t-1}^A + (1 - \beta_2)(\tilde{\nabla}_{A_t} L \odot \tilde{\nabla}_{A_t} L)$
            $A_{t+1} \leftarrow A_t - \eta \left(M_t^A \oslash (\sqrt{V_t^A} + \epsilon) + \gamma A_t\right)$
    **else**
        **Update** $B$**:**
            Only backpropagate w.r.t. $B_t$ and obtain $\nabla_{B_t} L$
            $\tilde{\nabla}_{B_t} L = \frac{1}{s^2} \nabla_{B_t} L (A_{t+1} A_{t+1}^\top)^{-1}$
            $\tilde{M}_t^B = M_{t-1}^B A_t A_{t+1}^\top (A_{t+1} A_{t+1}^\top)^{-1}$
            $M_t^B \leftarrow \beta_1 \tilde{M}_t^B + (1 - \beta_1) \tilde{\nabla}_{B_t} L$
            $V_t^B \leftarrow \beta_2 V_{t-1}^B + (1 - \beta_2)(\tilde{\nabla}_{B_t} L \odot \tilde{\nabla}_{B_t} L)$
            $B_{t+1} \leftarrow B_t - \eta \left(M_t^B \oslash (\sqrt{V_t^B} + \epsilon) + \gamma B_t\right)$

---

# D Proof and Details of Section 4

In this section, we will start to analyze the training paradigm of AltLoRA in Algorithm 1 and AltLoRA+ in Algorithm 2. In Appendix D.1, we first give the formal definition of stable feature learning in Definition 2. Then we will analyze our methods without momentum on a toy model in Appendix D.1.1. Furthermore, in Appendix D.1.2, we provably show that AltLoRA or AltLoRA+ with arbitrary LoRA ranks achieves stable feature learning in the infinite dimension NN setting. Then, in Appendix D.2, we provably show that AltLoRA would achieve transformation invariance. Finally, in Appendix D.3, within an over-parameterized two-layer ReLU NN tuning problem, we prove that AltLoRA or AltLoRA+ without momentum would converge linearly without the requirement of spectral initialization.

## D.1 Appendix for Section 4.1

First, let's recall the definition of stable feature learning below.

**Definition 2** (Stable Feature Learning (Definition A.1.[81])). *Consider any general LoRA layer* $BAx$ *with* $B \in \mathbb{R}^{k \times r}$ *and* $A \in \mathbb{R}^{r \times d}$ *being LoRA parameters. Denote* $\Delta_t = W_t - W_{t-1} = B_t A_t x - B_{t-1} A_{t-1} x$ *for fine-tuning step* $t$. *We say that LoRA mdoel achieves Stable Feature Learning when* $x, Ax, BAx \in \mathcal{O}(1)$ *for alll LoRA layers and* $\Delta_t \in \mathcal{O}(1)$ *for all fine-tuning step* $t$.

### D.1.1 Analysis on A Toy Model

Following LoRA+([21]), let's consider the simple linear model first

$$f(x) = (W + ba^T)x, \tag{25}$$

where $W \in \mathbb{R}^{1 \times n}$ is the pretrained model weight and $b \in \mathbb{R}$, $a \in \mathbb{R}^n$ are trainable LoRA parameters. Consider the quadratic loss function $\mathcal{L}(a, b) = (f(x) - y)^2/2$ with some scalar label $y$. We adopt Gaussian initialization $a \sim \mathcal{N}_n(0, \sigma^2 \mathbf{I}_n), b \sim \mathcal{N}(0, \sigma_b^2)$. Conventionally, $ba^T$ is initialized at zero for LoRA, and we thus consider setting $\sigma_a^2 = 0$, $\sigma_b^2 = \mathcal{O}(1)$.

For simplicity, assume AltLoRA or AltLoRA+ without momentum updates with learning rate $\eta = \mathcal{O}(n^c)$ for some $c \in \mathbb{R}$. Since the training process involves only elementary algebraic operations, the quantities there should be of powers of $n$. If we treat updates $A$ and $B$ each time as a single iteration, in iteration $t$, the feature update is given by

$$\begin{aligned}
\Delta f_{t+1} &:= f_{t+1}(x) - f_t(x) \\
&= \left( b_t a_t^T - \eta b_t (\tilde{\nabla}_{a_t} L)^T - \eta (\tilde{\nabla}_{b_t} L) a_{t+1}^T \right) x - b_t a_t^T x \\
&= -\eta (f_t(x) - y) \|x\|^2 - \eta (a_{t+1}^T x)^2 (f_{t+\frac{1}{2}}(x) - y) \|a_{t+1}\|^{-2},
\end{aligned}$$ (26)

where $f_{t+\frac{1}{2}}(x) := (W + b_t a_{t+1}^T) x$. We denote $\delta_t^1 = \eta b_t^2 (f_t(x) - y) \|x\|^2$, $\delta_t^2 = \eta (a_{t+1}^T x)^2 (f_{t+\frac{1}{2}}(x) - y)$. To achieve stable feature learning, it requires $\delta_t^1, \delta_t^2 \in \mathcal{O}(1)$ and further $f_t(x) \in \mathcal{O}(1) \ \forall t > 0$. Thus, we have the below modified linear constraints.

$$\begin{cases}
c + 1 = 0 & (\text{for } \delta_t^1 = \Theta(1)), \\
c + 2\gamma[a_{t+1}^T x] - \gamma[\|a_{t+1}\|^2] = 0 & (\text{for } \delta_t^2 = \Theta(1)), \\
\gamma[b_{t+1}] + \gamma[a_{t+1}^T x] = 0 & (\text{for } f_{t+1}(x) = \Theta(1)),
\end{cases}$$ (27)

where, for the sake of notational clarity, we introduce new notation $\gamma$ such that $v = \mathcal{O}(n^{\gamma[v]})$ captures the polynomial behavior for any $v$.

Solving the equations in (27), we can derive $c = -1$. With $\eta = \mathcal{O}(n^{-1})$, we get $\gamma[b_1] = \gamma[b_0] = 0$ and $\gamma[a_1^T x] = \gamma[\eta b_0^{-1} y \|x\|^2]$. Recursively, we can derive $b_t, a_t, \delta_t^1, \delta_t^2 \in \mathcal{O}(1)$ for all $t$. Therefore, we obtain $f_t \in \mathcal{O}(1)$ and $\Delta f_t \in \mathcal{O}(1)$. The above toy model illustrates that our proposed method achieve stable learning with learning rates for $A$ and $B$ of the same order of magnitude.

### D.1.2 Proof for Theorem 3

In this part, we extend the analysis above to a general neural architecture with LoRA layers. We show that the conclusion from the analysis on the linear model hold for general neural architecture.

**Assumption 1** (Assumption 1 in [21]). *We assume that the gradient processing step by AltLora in Algorithm 1 (or AltLoRA+ in Algorithm 2) satisfies $g_A^t = \mathcal{O}(n)$ for all $t$ where $g_A^t$ is the processed gradient of $A$ by AltLoRA (or AltLoRA+) in $t$-th update.*

**Lemma 2** (Lemma A.3. in [81]). *For any matrix $A \in \mathbb{R}^{m \times n}$, where $m$ being powers of $n$, such that $A^\top A$ is invertible and $\gamma[A_{ij}] = c$ for all $(i,j)$, we have*

$$\gamma \left[ (A^\top A)^{-1} \right] = -\gamma[\|a\|^2]$$

*with $a$ being any column of $A$.*

Now, we state the formal version of our Theorem 2.

**Theorem 5.** *Let $g_t^A$ and $g_t^B$ denote the processed gradient of $A$ and $B$, respectively, in Algorithm 1 or Algorithm 2. Assume Assumption 1 holds for the gradient processing of AltLoRA or AltLoRA+. And $g_t^A$ and $g_t^B \in \mathcal{O}(1)$ after the gradient processed. Further assume $BAx$ has dimension of $\mathcal{O}(n)$. Then the following results hold:*

*(1) AltLoRA (AltLoRA+) achieves stable feature learning with $\eta = \mathcal{O}(1)$.*

*(2) If we consider AltLoRA or AltLoRA+ without momentum, the update yields*

$$W_{t+1} = W_t - \eta Proj_{c(B_t)}(\nabla_{W_t} L) - \eta (\nabla_{W_{t+\frac{1}{2}}} L) Proj_{r(A_{t+1})},$$ (28)

*where $\eta Proj_{c(B_t)}(\nabla_{W_t} L), \eta (\nabla_{W_{t+\frac{1}{2}}} L) Proj_{r(A_{t+1})} \in \mathcal{O}(1)$. However, when doing joint update, the update will introduce additional across term $\eta^2 (\nabla_{W_t} L) A_t^T (A_t A_t^T)^{-1} (B_t^T B_t)^{-1} B_t^T (\nabla_{W_t} L) \in \mathcal{O}(1)$. The across term is indeed the second order term w.r.t $\eta$, but it is same magnitude as $\eta Proj_{c(B_t)}(\nabla_{W_t} L)$ and $\eta (\nabla_{W_t} L) Proj_{r(A)}$ in infinite-width NN setting.*

*Proof.* (**Part 1**) First, we will prove AltLoRA (AltLoRA+) can achieve stable feature learning. The technical lemmas and assumptions used for proof are also well-adapted in [21, 81].

We will alternately update $A$ first then update $B$. If we treat update $A$ frist then update $B$ as a single iteration, it could yield the update of the full model $W$ as

$$
\begin{aligned}
\Delta^t &= B_t A_t x - B_{t-1} A_{t-1} x \\
&= B_t A_t x - B_{t-1} A_t x + B_{t-1} A_t x - B_{t-1} A_{t-1} x \\
&= (B_t - B_{t-1}) A_t x + B_{t-1}(A_t - A_{t-1}) x \\
&= -\gamma g_B^{t-1}(A_t A_t^\top)^{-1} A_t x - \gamma B_{t-1}(B_{t-1}^\top B_{t-1})^\gamma g_A^{t-1} x.
\end{aligned}
\tag{29}
$$

Then we will denote these two parts of the update in the R.H.S of (29) as

$$
\begin{aligned}
\delta_1^t &= \eta B_{t-1}(B_{t-1}^\top B_{t-1})^\gamma g_A^{t-1} x \\
\delta_2^t &= \eta g_B^{t-1}(A_t A_t^\top)^{-1} A_t x.
\end{aligned}
\tag{30}
$$

Following Assumption 1, we know $g_A^{t-1} x \in \mathcal{O}(n)$. Thus the conditions of $\delta_1^t$, $\delta_2^t$, $B_{t-1} A_t x \in \mathcal{O}(x)$ are equivalent to

$$
\begin{aligned}
\gamma[\eta] + \gamma[B_{t-1}] + \gamma[(B_{t-1}^\top B_{t-1})^{-1}] + 1 &= 0 \\
\gamma[\eta] + \gamma[A_t A_t^\top] + \gamma[A_t x] &= 0.
\end{aligned}
\tag{31}
$$

For gradient update, we have

$$
\begin{aligned}
A_t x &= A_{t-1} x - \eta (B_{t-1}^\top B_{t-1})^{-1} g_A^{t-1} x \\
B_t &= B_{t-1} - \eta g_B^{t-1}(A_t A_t^\top)^{-1}.
\end{aligned}
\tag{32}
$$

thus we have

$$
\Rightarrow
\begin{cases}
\gamma[B_t] = \max\left\{\gamma[B_{t-1}], \gamma[\eta] + \gamma[(B_{t-1}^\top B_{t-1})^{-1}]\right\} \\
\gamma[A_t x] = \max\left\{\gamma[A_{t-1} x], \gamma[\eta] + \gamma[(B_{t[1}^\top B_{t-1})^{-1}] + 1\right\}.
\end{cases}
$$

Note $A_1 = A_0$, the recursive argument of $\delta_t^1$ and $\delta_t^2 \in \mathcal{O}(1)$ is the same as [81]. Therefore, we find that AltLoRA or AltLoRA+ achieves stable feature learning with $\eta = \mathcal{O}(1)$. We can conclude that our algorithm would achieve stable feature learning with the same order of $\eta$ in contrast to the standard LoRA ([21])

(**Part 2**) When removing the momentum in our methods, under Assumption 1, it would achieve stable feature learning as Part 1 has proved. Then the update of the full model $W$ is

$$
W_{t+1} = W_t - \eta Proj_{c(B_t)}(\nabla_{W_t} L) - \eta(\nabla_{W_{t+\frac{1}{2}}} L) Proj_{r(A_{t+1})},
\tag{33}
$$

where $\eta Proj_{c(B_t)}(\nabla_{W_t} L), \eta(\nabla_{W_{t+\frac{1}{2}}} L) Proj_{r(A_{t+1})} \in \mathcal{O}(1)$.

However, when doing a joint update with scaled gradient descent ([81]), the update of the full model $W$ is

$$
\begin{aligned}
W_{t+1} = {}& W_t - \eta Proj_{c(B_t)}(\nabla_{W_t} L) - \eta(\nabla_{W_t} L) Proj_{r(A_t)} \\
&+ \eta^2(\nabla_{W_t} L) A_t^T (A_t A_t^T)^{-1}(B_t^T B_t)^{-1} B_t^T(\nabla_{W_t} L)
\end{aligned}
\tag{34}
$$

where the additional cross term $\eta^2(\nabla_{W_t} L) A_t^T (A_t A_t^T)^{-1}(B_t^T B_t)^{-1} B_t^T(\nabla_{W_t} L)$ is of order $\mathcal{O}(1)$. While this term is second-order with respect to $\eta$, it shares the same magnitude as the first-order terms $\eta Proj_{c(B_t)}(\nabla_{W_t} L)$ and $\eta(\nabla_{W_t} L) Proj_{r(A_t)}$ under the infinite-width neural network setting. A straightforward explanation is that the embedding dimension contributes quadratically to the cross term's effect, matching the overall scale of the first two terms. □

## D.2 Proof of Section 4.2

First, let's restate Lemma 1 again and prove it.

**Lemma 3.** *If any two pairs of LoRA factors* $(A_1, B_1), (A_2, B_2)$ *satisfying*

$$W = W_0 + B_1 A_1 = W_0 + B_2 A_2, \tag{35}$$

*then*

$$\begin{aligned} Proj_{c(B_1)} &= Proj_{c(B_2)} \\ Proj_{r(A_1)} &= Proj_{r(A_2)} \end{aligned} \tag{36}$$

*where* $Proj_{c(\cdot)}$ *and* $Proj_{r(\cdot)}$ *is defined in (9).*

*Proof.* we know the column spaces of $B_1$ and $B_2$ are equivalent, as both of them span the column space of $W - W_0$. Thus, the projection matrices to the column spaces of $B_1$ and $B_2$ are the same, i.e., $Proj_{c(B_1)} = Proj_{c(B_2)}$, where $Proj_{c(\cdot)}$ is defined in (9). Similarly, the row spaces of $A_1$ and $A_2$ are equivalent. And the projection matrices to the column spaces of $A_1$ AND $A_2$ are the same, i.e., $Proj_{r(A_1)} = Proj_{r(A_2)}$. □

Lemma 1 tells that if two pairs of low-rank adaptation would get the same full model update, the projection matrix would preserve invariant over the pairs of low-rank adaptation. Next, we will restate Theorem 4 here and start to prove the theorem.

**Theorem 6.** *In Algorithm 1, every term is consistent across all equivalent LoRA pairs. Consequently, Algorithm 1 is transformation-invariant.*

*Proof.* Now we will use an inductive argument to prove it. Let's denote $(B_{1,t}, A_{1,t}), (B_{2,t}, A_{2,t})$ as two pairs of LoRA adaptation in the $t$-th interaction statisfying

$$W_0 + B_{1,t} A_{1,t} = W_0 + B_{2,t} A_{2,t}. \tag{37}$$

For the first pair $(B_{1,t}, A_{1,t})$, we denote $\tilde{M}_{1,t}^A$ and $M_{1,t}^A$ as the momentum used for $A_{1,t}$ in Algorithm 1. Let's assume, for the $(t-1)$-th iteration, we have the equivalent decomposition

$$B_{1,t-1} A_{1,t-1} = B_{1,t-1}, A_{1,t-1}. \tag{38}$$

Besides, we assume it is transformation invariance to $(t-1)$ iteration, then

$$B_{1,t-2} M_{1,t-2}^A = B_{2,t-2} M_{2,t-2}^A \tag{39}$$

$$M_{1,t-2}^B A_{1,t-2} = M_{1,t-2}^B A_{1,t-2}, \tag{40}$$

which implies that the historical information is invariant over the pairs of $(B_1, A_1)$ and $(B_2, A_2)$.

Then for the $t$-th iteration, we need to prove

$$B_{1,t-1} M_{1,t-1}^A = B_{2,t-1} M_{2,t-1}^A \tag{41}$$

$$M_{1,t-1}^B A_{1,t-1} = M_{2,t-1}^B A_{2,t-1}, \tag{42}$$

holds as well, and the update is transformation-invariant $B_{1,t} A_{1,t} = B_{2,t} A_{2,t}$.

First, we will focus on the update of $A$ and prove $B_{1,t-1} M_{1,t-1}^A = B_{2,t-1} M_{2,t-1}^A$. Recalling the definition of $M_{1,t}^A$ is the cumulative gradient to the time $t$ in Algorithm 1 , it yields

$$\begin{aligned} &B_{1,t-1} M_{1,t-1}^A \\ &= B_{1,t-1} \left( \beta_1 (B_{1,t-1}^\top B_{1,t-1})^{-1} B_{1,t-1}^T B_{1,t-2} M_{1,t-2}^A + (1-\beta_1) \frac{1}{s^2} (B_{1,t-1}^\top B_{1,t-1})^{-1} \nabla_{A_{1,t-1}} L \right) \\ &= \beta_1 Proj_{c(B_{1,t-1})} B_{1,t-2} M_{1,t-2}^A + (1-\beta_1) \frac{1}{s^2} B_{1,t-1} (B_{1,t-1}^\top B_{1,t-1})^{-1} \nabla_{A_{1,t-1}} L \\ &= \beta_1 Proj_{c(B_{1,t-1})} B_{1,t-2} M_{1,t-2}^A + (1-\beta_1) \frac{1}{s} Proj_{c(B_{1,t-1})} \nabla_{W_{1,t-1}} L, \end{aligned} \tag{43}$$

where the last line uses the results in (1) and $W_{1,t-1} := W_0 + B_{1,t-1}A_{1,t-1}$. Next, under the assumption for induction in (41) and Lemma 1, it yields

$$
\begin{aligned}
B_{1,t-1}M_{1,t-1}^A &= \beta_1 Proj_{c(B_{1,t-1})} B_{1,t-2}M_{1,t-2}^A + (1-\beta_1)\frac{1}{s}Proj_{c(B_{1,t-1})}\nabla_{W_{1,t-1}}L \\
&= \beta_1 Proj_{c(B_{2,t-1})} B_{2,t-2}M_{2,t-2}^A + (1-\beta_1)\frac{1}{s}Proj_{c(B_{2,t-1})}\nabla_{W_{2,t-1}}L \quad (44)\\
&= B_{2,t-1}M_{2,t-1}^A.
\end{aligned}
$$

After updating $A$, we can find the update of the full model as

$$
\begin{aligned}
B_{1,t-1}A_{1,t} &= B_{1,t-1}(A_{1,t-1} - \eta M_{1,t-1}^A) \\
&= B_{1,t-1}A_{1,t-1} - \eta B_{1,t-1}M_{1,t-1}^A \\
&= B_{2,t-1}A_{2,t-1} - \eta B_{2,t-1}M_{2,t-1}^A \quad (45)\\
&= B_{2,t-1}A_{2,t},
\end{aligned}
$$

where the second-to-last line uses the results (38) in $(t-1)$-th iteration and the results in (44). Again, reapplying Lemma 1, we can find that $Proj_{c(A_{1,t})} = Proj_{c(A_{2,t})}$.

Up to now, we have shown that the update of $A$ is transformation-invariant and $B_{1,t-1}M_{1,t-1}^A = B_{1,t-1}M_{1,t-1}^A$. With a similar argument, we can prove $M_{1,t-1}^B A_{1,t-1} = M_{1,t-1}^B A_{1,t-1}$ and $B_{1,t}A_{1,t} = B_{2,t}A_{2,t}$. Therefore, with the inductive argument, we prove the update of Algorithm 1 is transformation-invariant. □

In contrast to the prior work [79], our analysis centers on Lemma 1 to establish the proof of Theorem 4. Leveraging the alternating update strategy in Algorithm 1, we analyze the contributions of $A$ and $B$ to the full model update separately, allowing us to rigorously demonstrate transformation invariance. In comparison, [79] adopts a joint update of $A$ and $B$, which introduces a cross term $\delta B \delta A$ that is ignored in their analysis, resulting in an inexact form of transformation invariance. Our alternating approach provides a principled direction toward achieving exact transformation invariance.

**Discussion** With our newly designed momentum mechanism, the first-order momentum terms remain consistent across all equivalent LoRA pairs, thereby ensuring that AltLoRA is robust to transformation invariance. In contrast, AltLoRA+ does not preserve this invariance. Motivated by this observation, we further attempt to design a second-order momentum mechanism that aligns optimally within the low-rank space under memory constraints. Although the second-order momentum terms are individually consistent across equivalent LoRA pairs, their combination with the first-order momentum leads to inconsistencies, ultimately breaking transformation invariance. To address this issue, employing unscaled gradients and momentum, as demonstrated by LoRA-Rite [79], could be a viable solution. However, as this approach diverges from our primary focus, we leave it for future work.

### D.3 Convergence Analysis

#### D.3.1 Set Up

Following the previous work ([81]), we provide a convergence analysis of the proposed algorithm within the over-parameterized two-layer ReLU NN tuning problem. For a data matrix $X \in \mathbb{R}^{n \times d}$ and and any arbitrary vector $u$, we consider a set of diagonal matrices $\{\text{diag}([Xu \geq 0]) \mid u \in \mathbb{R}^d\}$, which take value 1 or 0 along the diagonals that indicate the set of possible arrangement activation patterns for the ReLU activation. Let the distinct elements of this set be denoted as $D_1, \ldots, D_P$ (see [81] for more details). The constant $P$ corresponds to the total number of partitions of $\mathbb{R}^d$ by hyperplanes passing through the origin that are also perpendicular to the rows of $X$ [49]. Intuitively, $P$ can be regarded as the number of possible ReLU activation patterns associated with $X$. [49] explains that a two-layer ReLU problem shares the same optimal objective with the convex problem

$$
\min_{W_i \; i \in [P]} \frac{1}{2} \left\| \sum_{i=1}^{P} D_i X W_i - Y \right\|_F^2. \quad (46)
$$

As we focus on fine-tuning, given a pretrained model with model weights $\{W_i\}_{i=1}^P$, we can do low-rank adaptation and rewrite the problem (46) as

$$\min_{A_i, B_i, i=1, \cdots P} \frac{1}{2} \left\| \sum_{i=1}^P D_i X (W_i + B_i A_i) - Y \right\|_F^2, \tag{47}$$

where $X \in \mathbb{R}^{n \times d}$, $A_i \in \mathbb{R}^{r \times c}$, $B_i \in \mathbb{R}^{d \times r}$ and $Y \in \mathbb{R}^{n \times c}$. We consider the response model $Y = \sum_i^P D_i X (W_i + B_i^\star A_i^\star)$. We define $X^\star := \sum_i^P B_i^\star A_i^\star$ are fixed and unknown matrices. Let's denote $\sigma_r(\cdot)$ as the $r$-th largest singular value. First let's introduce the definition of Restricted Isometry Property (RIP).

**Definition 3.** *(Restricted Isometry Property, [53]) The matric $C \in \mathbb{R}^{n \times d}$ is said to satisfy Restricted Isometry Property(RIP) with parameters $(r, \delta_r)$ if there exists constants $0 \le \delta_r \le 1$, for any matrices $M \in \mathbb{R}^{d \times c}$ with rank $r$, the below holds*

$$(1 - \delta_r) \|M\|_F^2 \le \|CM\|_F^2 \le (1 + \delta_r) \|M\|_F^2. \tag{48}$$

RIP is a widely used condition in the filed of compressed sensing ([42, 15, 53, 73]), which states that the operator $C$ approximately preserves distances between low-rank matrices. In the absence of noise, we can establish a direct relationship between the loss function and the recovery error. If we denote $C_i := D_i X$, Problem (47) is equivalent to the problem below up to a change of labels

$$\min_{A_i, B_i, i=1, \cdots P} L_c(\boldsymbol{B}, \boldsymbol{A}) := \frac{1}{2} \left\| \sum_i^P C_i (B_i A_i - X_\star) \right\|_F^2, \tag{49}$$

where $\boldsymbol{B} = \{B_1, \cdots, B_P\}$ and $\boldsymbol{A} = \{A_1, \cdots, A_P\}$.

**Notation** Inspired by the previous work [42, 83, 84], we introduce two local norms and their corresponding dual norms for a matrix $W \in \mathbb{R}^{k \times r}$

$$P_{A_t^i} := A_t^i (A_t^i)^T, \quad \|W\|_{P_{A_t^i}} := \|W P_{A_t^i}^{\frac{1}{2}}\|_F, \quad \|W\|_{P_{A_t^i}^\star} := \|W P_{A_t^i}^{-\frac{1}{2}}\|_F,$$

$$P_{B_t^i} := (B_t^i)^T B_t^i, \quad \|W\|_{P_{B_t^i}} := \|W P_{B_t^i}^{\frac{1}{2}}\|_F, \quad \|W\|_{P_{A_t^i}^\star} := \|W P_{B_t^i}^{-\frac{1}{2}}\|_F. \tag{50}$$

Here, we assume $A_t^i$ and $B_t^i$ are of full rank $r$ for any $i$. If they aren't of full rank, we can replace them with the Moore-Penrose inverse([4]). Now we are ready to establish the convergence analysis.

### D.3.2 Useful Lemma

For the $t$-th iteration, let's denote $\boldsymbol{B}_t = \{B_t^1, \cdots, B_t^P\}$ and $\boldsymbol{A}_t = \{A_t^1, \cdots, A_t^P\}$. If we apply AltLoRA or AltLoRA+ without momentum for Problem (49), for any $i \in [P]$, the alternating update rule as we proposed can be written as

$$A_{t+1}^i \leftarrow A_t^i - \eta (B_t^i (B_t^i)^T)^{-1} \nabla_{A_t^i} L_c(\boldsymbol{B}_t, \boldsymbol{A}_t)$$

$$B_{t+1}^i \leftarrow B_t^i - \eta \nabla_{B_t^i} L_c(\boldsymbol{B}_t, \boldsymbol{A}_{t+1}) ((A_{t+1}^i)^T A_{t+1}^i)^{-1}. \tag{51}$$

First, we will list some assumptions used in our analysis.

**Assumption 2.** *Suppose that $C_i = D_i X$ obeys the $r$-RIP with a constant $\delta_r$ for each $i$.*

**Assumption 3.** *Suppose that $\|C_i^T C_j\|_2 := \|X^T D_i^T D_j X\|_2 \le \frac{1 + \delta_r}{P(P-1)}$*

Assumption 2 and 3 also adopt in [81] to analyze their optimizer for LoRA. For matrix $X$ with $i.i.d$ Gaussian entries $\mathcal{N}(0, 1/d \|D_i\|_0)$, $D_i X$ satisfies RIP for a constant $\delta_r$ when $\|D_i\|_0$ is on the order of $r(d+c)/(d\delta_r^2)$. Note $\|X^T D_i^T D_j X\|_2 \le \|X^T X\|_2$ for all $(i,j)'s$. Thus bounding $\|X^T D_i^T D_j X\|_2$ amounts to bounding the largest singular value of the empirical covariance.

**Lemma 4.** *For a given $i \in [P]$, the gradient of Problem (49) are*

$$\nabla_{A_t^i} L(\boldsymbol{B}, \boldsymbol{A}) = \sum_j^P (B_t^i)^T (C_i)^T C_j (B_t^j A_t^j - X_\star)$$

$$\nabla_{B_t^i} L(\boldsymbol{B}, \boldsymbol{A}) = \sum_j^P (C_i)^T C_j (B_t^j A_{t+1}^j - X_\star) (A_{t+1}^j)^T. \tag{52}$$

*Proof.* For any given $i$ and $t$, it yields

$$\nabla_{A_t^i} L(\boldsymbol{B}, \boldsymbol{A}) = \frac{\partial}{\partial A_t^i} \left\{ \frac{1}{2} \left\| \sum_j^P C_j(B_j A_j - X_\star) \right\|_F^2 \right\} = \sum_j^P (B_t^i)^T (C_i)^T C_j (B_t^j A_t^j - X_\star). \quad (53)$$

Similarly, we can derive the $\nabla_{B_t^i} L(\boldsymbol{B}, \boldsymbol{A})$ as shown in (52). $\qquad\square$

**Lemma 5.** *Suppose Assumption 2 and 3 holds, then we have*

$$L_c(\boldsymbol{B}_t, \boldsymbol{A}_{t+1}) \leq L_c(\boldsymbol{B}_t, \boldsymbol{A}_t) - c_1 \max_i \left\| \nabla_{A_t^i} L_c(\boldsymbol{B}_t, \boldsymbol{A}_t) \right\|_{P_{B_t^i}^\star}^2$$

$$\qquad (54)$$

$$L_c(\boldsymbol{B}_{t+1}, \boldsymbol{A}_{t+1}) \leq L_c(\boldsymbol{B}_t, \boldsymbol{A}_{t+1}) - c_1 \max_i \left\| \nabla_{B_t^i} L_c(\boldsymbol{B}_t, \boldsymbol{A}_{t+1}) \right\|_{P_{A_{t+1}^i}^\star}^2,$$

*where $c_1 = P(\eta - \frac{\eta^2(1+\delta_r+\frac{1}{P})}{2})$.*

*Proof.* Using the update rule in (51), we have

$$L_c(\boldsymbol{B}_t, \boldsymbol{A}_{t+1}) = \frac{1}{2} \left\| \sum_i^P C_i(B_t^i A_{t+1}^i - X_\star) \right\|_F^2$$

$$= \frac{1}{2} \left\| \sum_i^P C_i \left( B_t^i \left( A_t^i - \eta((B_t^i)^T B_t^i)^{-1} \nabla_{A_t^i} L_c(\boldsymbol{B}_t, \boldsymbol{A}_t) \right) - X_\star \right) \right\|_F^2$$

$$= \frac{1}{2} \left\| \sum_i^P C_i(B_t^i A_t^i - X_\star) \right\|_F^2 \qquad (55)$$

$$+ \underbrace{\frac{\eta^2}{2} \left\| \sum_i^P C_i B_t^i((B_t^i)^T B_t^i)^{-1} \nabla_{A_t^i} L_c(\boldsymbol{B}_t, \boldsymbol{A}_t) \right\|_2^2}_{T_1}$$

$$- \eta \underbrace{\left\langle \sum_i^P C_i(B_t^i A_t^i - X_\star), \sum_i^P C_i B_t^i((B_t^i)^T B_t^i)^{-1} \nabla_{A_t^i} L_c(\boldsymbol{B}_t, \boldsymbol{A}_t) \right\rangle}_{T_2}$$

For $T_1$, recalling Lemma 4, then we have

$$T_1 \leq \frac{\eta^2}{2} \sum_i^P \| C_i B_t^i((B_t^i)^T B_t^i)^{-1} \nabla_{A_t^i} L_c(\boldsymbol{B}_t, \boldsymbol{A}_t) \|_F^2$$

$$+ \frac{\eta^2}{2} \sum_{i \neq j} \left\langle C_i B_t^i((B_t^i)^T B_t^i)^{-1} \nabla_{A_t^i} L_c(\boldsymbol{B}_t, \boldsymbol{A}_t), C_j B_t^j((B_t^j)^T B_t^j)^{-1} \nabla_{A_t^j} L_c(\boldsymbol{B}_t, \boldsymbol{A}_t) \right\rangle$$

$$\overset{(a)}{\leq} \frac{\eta^2(1+\delta_r)}{2} P \max_i \| \nabla_{A_t^i} L_c(\boldsymbol{B}_t, \boldsymbol{A}_t) \|_{P_{B_t^i}^\star}^2 \qquad (56)$$

$$+ \frac{\eta^2}{2} \max_{i \neq j} \| C_i^T C j \|_2 P(P-1) \max_i \| \nabla_{A_t^i} L_c(\boldsymbol{B}_t, \boldsymbol{A}_t) \|_{P_{B_t^i}}^2$$

$$\overset{(b)}{\leq} \frac{\eta^2(1+\delta_r+\frac{1}{P})}{2} P \max_i \| \nabla_{A_t^i} L_c(\boldsymbol{B}_t, \boldsymbol{A}_t) \|_{P_{B_t^i}}^2,$$

where (a) uses Cauchy Inequality, Assumption 2 and the fact that $\| B_t^i((B_t^i)^T B_t^i)^{-\frac{1}{2}} \|_2^2 = 1$, (b) uses the assumption that $\max_{i \neq j} \| C_j^T C_j \|_2 \leq \frac{(1+\delta_r)}{P(P-1)}$.

For $T_2$, using Lemma 4 again, we have

$$
\begin{aligned}
T_2 &= \eta \left\langle \sum_j^P C_j(B_t^j A_t^j - X_\star), \sum_j^P C_j B_t^j ((B_t^j)^T B_t^j)^{-1} \nabla_{A_t^j} L_c(\boldsymbol{B}_t, \boldsymbol{A}_t) \right\rangle \\
&= \eta \sum_j^P \left\langle \sum_i^P C_i(B_t^i A_t^i - X_\star), C_j B_t^j ((B_t^j)^T B_t^j)^{-1} \nabla_{A_t^j} L_c(\boldsymbol{B}_t, \boldsymbol{A}_t) \right\rangle \\
&= \eta \sum_i^P \left\| \nabla_{A_t^i} L_c(\boldsymbol{B_t}, \boldsymbol{A}_t) \right\|_{P_{B_t^i}^\star}^2 \\
&\le \eta P \max_i \| \nabla_{A_t^i} L_c(\boldsymbol{B_t}, \boldsymbol{A}_t) \|_{P_{B_t^i}^\star}^2 .
\end{aligned}
\tag{57}
$$

To sum up, it yields

$$
L_c(\boldsymbol{B}_t, \boldsymbol{A}_{t+1}) \le L_c(\boldsymbol{B}_t, \boldsymbol{A}_t) - \left( \eta - \frac{\eta^2(1 + \delta_r + \frac{1}{P})}{2} \right) P \max_i \left\| \nabla_{A_t^i} L_c(\boldsymbol{B_t}, \boldsymbol{A}_t) \right\|_{P_{B_t^i}^\star}^2 .
\tag{58}
$$

Similarly, we can induce

$$
L_c(\boldsymbol{B}_{t+1}, \boldsymbol{A}_{t+1}) \le L_c(\boldsymbol{B}_t, \boldsymbol{A}_{t+1}) - \left( \eta - \frac{\eta^2(1 + \delta_r + \frac{1}{P})}{2} \right) P \max_i \left\| \nabla_{B_t^i} L_c(\boldsymbol{B}_t, \boldsymbol{A}_{t+1}) \right\|_{P_{A_{t+1}^i}^\star}^2 .
\tag{59}
$$

$\square$

**Lemma 6.** *Suppose Assumption 2 holds, then, for any $i \in [P]$, we have*

$$
\begin{aligned}
\| \nabla_{A_t^i} L_c(\boldsymbol{B}_t, \boldsymbol{A}_t) \|_{P_{B_t^i}^\star}^2 &\ge 2(1 - \delta_r) L_c(\boldsymbol{B}_t, \boldsymbol{A}_t) \\
\| \nabla_{B_t^i} L_c(\boldsymbol{B}_t, \boldsymbol{A}_{t+1}) \|_{P_{A_{t+1}^i}^\star}^2 &\ge 2(1 - \delta_r) L_c(\boldsymbol{B}_t, \boldsymbol{A}_{t+1}) .
\end{aligned}
\tag{60}
$$

*Proof.* See Lemma 6 in [42] for the detailed proof. $\square$

**Theorem 7.** *Assume for any $i \in [p]$ the matrix $C_i = D_i X$ satisfies the rank $r$-RIP with constant $\delta_r$ (Assumption 2) and $0 \le \eta \le \frac{1}{1 + \delta_r + \frac{1}{P}}$, then AltLoRA or AltLoRA+ without momentum solves the over-parameterized problem leads to*

$$
L_c(\boldsymbol{B}_{t+1}, \boldsymbol{A}_{t+1}) \le (1 - \eta_c)^2 L_c(\boldsymbol{B}_t, \boldsymbol{A}_t)
\tag{61}
$$

*and*

$$
\left\| \sum_i^P B_t^i A_t^i - X_\star \right\|_F^2 \le \frac{1 + \delta_r}{1 - \delta_r} (1 - \eta_c)^{2t} \left\| \sum_i^P B_0^i A_0^i - X_\star \right\|_F^2 ,
\tag{62}
$$

*where $\eta_c = 2P(1 - \delta_r) \left( \eta - \frac{\eta^2(1 + \delta_r + \frac{1}{P})}{2} \right)$.*

*Proof.*

$$
\begin{aligned}
L_c(\boldsymbol{B}_{t+1}, \boldsymbol{A}_{t+1}) &\le L_c(\boldsymbol{B}_t, \boldsymbol{A}_{t+1}) - \left( \eta - \frac{\eta^2(1 + \delta_r + \frac{1}{P})}{2} \right) P \max_i \left\| \nabla_{B_t^i} L_c(\boldsymbol{B}_t, \boldsymbol{A}_{t+1}) \right\|_{P_{A_{t+1}^i}^\star}^2 \\
&\le L_c(\boldsymbol{B}_t, \boldsymbol{A}_{t+1}) - \left( \eta - \frac{\eta^2(1 + \delta_r + \frac{1}{P})}{2} \right) 2P(1 - \delta_r) L_c(\boldsymbol{B}_t, \boldsymbol{A}_{t+1}) \\
&\le \left( 1 - 2P(1 - \delta_r) \left( \eta - \frac{\eta^2(1 + \delta_r + \frac{1}{P})}{2} \right) \right) L_c(\boldsymbol{B}_t, \boldsymbol{A}_{t+1}) \\
&\le (1 - \eta_c)^2 L_c(\boldsymbol{B}_t, \boldsymbol{A}_t),
\end{aligned}
\tag{63}
$$

where we apply Lemma 5 and 6 and $\eta_c = 2P(1 - \delta_r)\left(\eta - \frac{\eta^2(1+\delta_r+\frac{1}{P})}{2}\right)$. Moreover, under Assumption 2, we have

$$\left\|\sum_i^P B_t^i A_t^i - X_\star\right\|_F^2 \leq \frac{1+\delta_r}{1-\delta_r}(1-\eta_c)^{2t}\left\|\sum_i^P B_0^i A_0^i - X_\star\right\|_F^2.\tag{64}$$

$\square$

# E  Appendix for Expirments

## E.1  Details and Results for Supervised Fine-tuning

For the experimental setup, we follow the configuration used in LoRA-Pro [66] and summarize the key description here. As the experiments involve randomness from initialization and optimization, all results are averaged over three different random seeds.

**Dialogue Generation Task** We fine-tune large language models on a 52k subset of the WizardLM dataset [72] and evaluate it using the MT-Bench dataset [89]. GPT-4o is used to asses the quality of the model's response and we report the first-turn score as the metric.

**Math Task** We fine-tuning large language models on a 100k sample from the MetaMathQA dataset [80]. The model is then evaluated on the GSM8K test set [11], and we report the accuracy as the metric.

**Coding Task** We fine-tuning large language models on a 100k subset of the CodeFeedBack dataset [90] and test it on the HumanEval dataset [6], reporting the PASS@1 metric.

For the choice of learning rate, we perform grid search for LoRA, its variants, and AltLoRA+ over 1e-5, 4e-5, 1e-4. Since AltLoRA does not use second-moment estimates, we conduct an extended grid search over 1e-2, 1e-3, 1e-4, 4e-5, 1e-5. We observe that AltLoRA performs better with higher learning rates, and therefore report results using 1e-2, 1e-3, 1e-4 in the main evaluation. We set the iteration number to be 1 and the max step is 3000 for each experiment.

## E.2  Additional Results

In Table 5, we compare our method with existing approaches across the three tasks described on Llama-3-8B model. Our method further bridges the performance gap between LoRA and full fine-tuning.

Table 5: Comparison of different LoRA variants on MT-Bench, GSM8K, and HumanEval benchmarks (accuracy in %) on Llama-3-8B-Base.

| Method | MT-Bench | GSM8K | HumanEval |
|---------|----------|-------|-----------|
| PreTrain | 5.63 | 49.96± 0.38 | 34.76±0.37 |
| LoRA | 6.20 | 62.11±0.13 | 37.71±0.12 |
| AltLoRA | 6.05 | 64.39±0.23 | 40.81±0.47 |
| AltLoRA+ | 6.34 | 67.38±0.13 | 43.81±0.31 |

In Table 6, we report the training loss of AltLoRA and its baselines on GSM8K using Llama-3-8B-Base model. Our method not only converges to a lower training loss than full fine-tuning, but also does so more rapidly than most baselines.

Base Llama-3-8B base and Llama-3.1-8B base model, we also consider the instruct tuning model-Llama3-8B-Instruct model. We fine-tune it on the MetaMathQA dataset and evaluate it on the GSM8K test set, reporting accuracy as the primary metric. In Table 7, our results demonstrate that even when using a stronger pretrained model, low-rank adaptation can still yield performance gains. More importantly, both AltLoRA and AltLoRA+ consistently outperform the standard LoRA baseline. We will add the full comparison in the revised revision.

Table 6: Training loss on GSM8K (Llama-3.1-8B) across training steps (lower is better).

| Method | step 500 | step 1000 | step 1500 | step 2000 | step 2500 | step 3000 |
|---|---|---|---|---|---|---|
| Full Fine-tuning | 0.51 | 0.28 | 0.24 | 0.18 | 0.18 | 0.18 |
| LoRA | 0.57 | 0.30 | 0.26 | 0.23 | 0.22 | 0.22 |
| LoRA-Pro | 0.51 | 0.31 | 0.24 | 0.22 | 0.21 | 0.21 |
| LoRA-GA | 0.45 | 0.24 | 0.20 | 0.18 | 0.18 | 0.18 |
| LoRA-Rite | 0.55 | 0.31 | 0.24 | 0.22 | 0.21 | 0.21 |
| AltLoRA | 0.48 | 0.25 | 0.21 | 0.19 | 0.18 | 0.18 |
| AltLoRA+ | 0.45 | 0.23 | 0.19 | 0.18 | 0.17 | 0.17 |

Table 7: GSM8K accuracy (%) on Llama3-8B-instruct and its fine-tuned variants.

| Method | GSM8K |
|---|---|
| Llama3-8B-instruct | 61.24 |
| + *Full FT* | 83.26 |
| + *LoRA* | 81.12 |
| + *AltLoRA* | 82.60 |
| + *AltLoRA+* | 83.88 |

### E.3 Additional Ablation Study

We conduct additional ablation studies to further demonstrate the practical effectiveness of our proposed methods. In Appendix E.3.1, we evaluate the performance of our methods under varying hyperparameter settings on the LLaMA 3-8B model. Furthermore, in Appendix E.3.2, beyond the learning rate, scaling factor $\alpha$, and rank examined in Table 1, we perform comprehensive ablation studies for both AltLoRA and AltLoRA+ on the LLaMA 3.1-8B model.

### E.3.1 Additional Ablation Study for Llama-3-8B Model

We further conduct ablation studies on the LLaMA 3-8B model to evaluate the robustness of our method under varying hyperparameter settings. As shown in Figure 2, we compare the performance of LoRA, AltLoRA, and AltLoRA+ on the GSM8K dataset across different learning rates and scaling factors $\alpha \in \{8, 16, 32\}$. AltLoRA+ consistently outperforms the baselines across all configurations, demonstrating both higher accuracy and stronger robustness to hyperparameter variation. We also have that all methods have better performance using $\alpha = 16$.

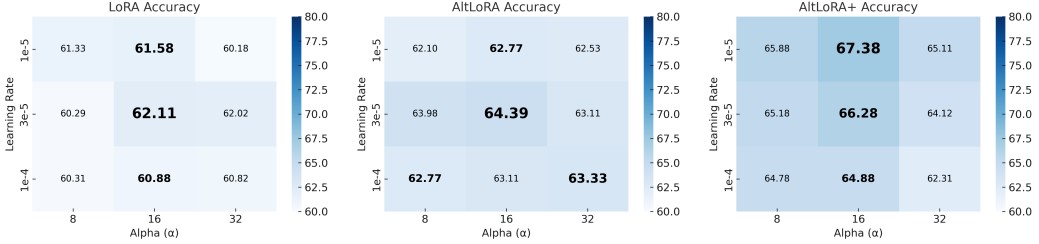

Figure 2: Evaluation Accuracy of LoRA, AltLoRA and AltLoRA+ for various learning rate $\eta$ and scaling factor $\alpha$ combination on the GSM9K using Llama-3-8B.

### E.3.2 Additional Ablation Study for Llama 3.1-8B Model

**Training Epoch** To rule out under-training as a potential factor for empirical performance, we increase the training budget to *5 epochs* under the same setup as Table 2. Results are reported in Table 8. (i) More epochs generally improve all methods; (ii) *AltLoRA* and *AltLoRA+* retain and even strengthen their advantage—*AltLoRA+* reaches $\sim$76% on GSM8K and approaches Full FT on

Table 8: Comparison of different LoRA variants on MT-Bench, GSM8K, and HumanEval benchmarks on Llama-3.1-8B-Base (5 training epoches). **Bold** indicates the best result, underline represents the second-best one.

| Method | MT-Bench | GSM8K | HumanEval |
|---|---|---|---|
| Full FT | 6.24±0.03 | 75.31±0.47 | **50.42±1.20** |
| LoRA | 6.19±0.02 | 66.78±1.14 | 41.54±1.01 |
| PiSSA | 6.11±0.03 | 67.39±0.61 | 42.12±1.45 |
| rsLoRA | 6.18±0.02 | 67.98±1.21 | 44.31±1.31 |
| LoRA+ | 6.21±0.02 | 74.00±0.61 | 44.32±1.33 |
| DoRA | 6.23±0.03 | 67.12±0.84 | 45.12±1.32 |
| AdaLoRA | 6.05±0.02 | 68.29±0.98 | 42.58±1.84 |
| LoRA-GA | 6.24±0.05 | 73.44±1.10 | 45.56±1.48 |
| LoRA-Pro | 6.22±0.02 | 73.24±0.79 | 43.21±1.50 |
| LoRA-Rite | 6.20±0.03 | 73.48±0.65 | 45.46±1.21 |
| AltLoRA | **6.26±0.04** | 74.52±0.37 | 46.28±1.11 |
| AltLoRA+ | 6.18±0.04 | **76.01±0.34** | 50.01±1.01 |

HumanEval; (iii) compared with standard LoRA and its variants (LoRA-Pro, LoRA-Rite, LoRA-GA), our methods benefit more consistently as the training budget grows, indicating that the gains are not a short-iteration artifact but persist under stronger training.

**Ablation study on the updating strategy** In Table 9, in contrast to joint update with scaled gradient descent [81], AltLoRA can optimally approximates the full gradient with alternating update and obtain better performance in evaluation. Interestingly, we find that the alternating update scheme—where matrix $B$ is updated before $A$—consistently yields better performance. One possible explanation is that, under the standard initialization where $B$ is set to zero, updating $A$ first does not lead to meaningful descent.

Table 9: Performance comparison of LoRA, AltLoRA and AltLoRA+ on the GSM8K and Llama 3.1 8B with different updating strategies.

| GSM8K | LoRA | AltLoRA | AltLoRA+ |
|---|---|---|---|
| Alternating ($A$ first) | 66.11 | 74.49 | 76.91 |
| Alternating ($B$ first) | 67.66 | 76.31 | 76.97 |
| Joint Update | 66.43 | 74.21 | 76.56 |

**Ablation study on the number of LoRAs** As low-rank adaptation comes to be a popular parameter-efficient technique for fine-tuning, it's well applied to more complicated scenarios ([43, 77, 57, 23, 70]). Notably, a very significant application is to improve the structure of the mixture of experts with parameter efficiency([70, 37]), handling multiple tasks simultaneously ([48, 26]) and addressing catastrophic forgetting ([14]). Following the work ([70]), we explore the performance as the number of LoRAs varies and utilize the gating balancing loss. Additionally, we compare AltLoRA and standard LoRA on the GSM8K dataset using the Llama 3.1-8B model(see Table 10). In our experiments, the number of LoRA experts is set to $\{1, 4, 8\}$, and the entropy regularization weight is 0.0001. We observe that increasing the number of LoRA experts enhances the capacity of the language model, leading to improved performance.

Table 10: Comparison of the mixture of experts model, with different expert numbers on GSM8K and Llama 3.1-8B-Base

| Expert Num | LoRA | AdaLoRA | LoRA+ | AltLoRA | AltLoRA+ |
|---|---|---|---|---|---|
| 1 | 66.11 | 72.63 | 72.33 | 74.49 | 76.91 |
| 4 | 67.43 | 71.71 | 71.27 | 75.01 | 77.33 |
| 8 | 67.89 | 70.34 | 71.44 | 75.33 | 76.94 |

**Ablation Study on Initialization.** To further validate the robustness of our method with respect to initialization, as discussed in Section 4.3, we conduct an ablation study using different initialization strategies. "Gaussian" refers to the standard random initialization used in the original LoRA framework [25]. "Kaiming" denotes the widely adopted Kaiming initialization, which is designed to maintain variance stability across layers. "Spectral" represents an initialization strategy based on spectral decomposition, where we perform singular value decomposition (SVD) on the pretrained weight matrix and construct the low-rank components using the top-$r$ singular vectors, like the initialization proposed in [88]. In Table 11, we can see that with different initialization strategies, our method would achieve a superior performance over the standard LoRA. Without spectral initialization, using Kaiming initialization for $A$ and setting $B$ to be zero would achieve the best performance. Besides, to ensure the initial update of $BA$ is zero, one of the matrices must be initialized to zero. Notably, setting $B = 0$ while using a small initialization for $A$ yields better performance compared to the reverse setup. This finding is consistent with observations in existing literature [20].

Table 11: Comparison of the initialization strategies on GSM8K and Llama 3.1-8B-Base

| Initialization Strategy | | LoRA | AltLoRA | AltLoRA+ |
|---|---|---|---|---|
| **A** | **B** | | | |
| Gaussian | zero | 66.37 | 73.13 | 76.87 |
| zero | Gaussian | 66.18 | 72.13 | 76.50 |
| Kaiming | zero | 65.11 | 74.49 | 76.91 |
| zero | Kaiming | 67.10 | 74.03 | 76.88 |
| Spectral | zero | 67.63 | 74.67 | 76.60 |
| zero | Spectral | 67.10 | 74.61 | 76.37 |

## E.4 The performance and efficiency trade-off

We summarize the memory–accuracy trade-off of our method against strong LoRA baselines on GSM8K with the Llama-3.1-8B backbone in Table 12. Under comparable memory budgets, **AltLoRA** consistently improves accuracy over standard LoRA and LoRA-Rite, and **AltLoRA+** achieves the best overall accuracy while maintaining a memory footprint close to standard LoRA. This demonstrates that our optimization yields a more favorable efficiency frontier: for a given memory cost, AltLoRA variants deliver higher accuracy; for a target accuracy, they require less memory.

Table 12: Memory cost (GB) vs. accuracy (%) on GSM8K (fine-tuning Llama-3.1-8B). AltLoRA and AltLoRA+ improve accuracy while preserving LoRA-level memory efficiency.

| Method | Memory Cost (GB) | Accuracy (%) |
|---|---|---|
| Full FT | $> 48$ | 73.31 |
| LoRA | 22.26 | 66.11 |
| LoRA-Rite | 25.39 | 74.10 |
| LoRA-Pro | 40.12 | 73.12 |
| **AltLoRA** | **22.56** | **74.49** |
| **AltLoRA+** | **23.16** | **76.91** |

