# OpenReview forum: "AltLoRA: Towards Better Gradient Approximation in Low-Rank Adaptation with Alternating Projections"
_NeurIPS.cc/2025/Conference — NeurIPS 2025 poster_

### Official Review · Reviewer_uqXj · 2025-06-23

**Clarity:** 3
**Significance:** 2
**Originality:** 2
**Rating:** 4
**Confidence:** 3

**Summary:**

The paper proposes AltLoRA, a novel method for Low-Rank Adaptation (LoRA) that aims to bridge the performance gap between LoRA and full fine-tuning by introducing an alternating projection strategy for gradient approximation and momentum optimization within low-rank constraints. The authors provide theoretical guarantees for convergence, stable feature learning, and transformation invariance, supported by extensive experiments across multiple tasks (natural language understanding, dialogue generation, mathematical reasoning, and code generation). The method claims to outperform existing LoRA variants while maintaining memory efficiency.

**Questions:**

1. Could the authors elaborate on why second-order momentum breaks transformation invariance? Is there a principled way to align it with low-rank subspaces, or is this a fundamental limitation?

**Ethical Concerns:**

["NO or VERY MINOR ethics concerns only"]

**Final Justification:**

I consider maintaining my score.

**Quality:**

3

**Strengths And Weaknesses:**

- Strengths
1. The alternating projection approach for gradient approximation and momentum design is a fresh take on improving LoRA, addressing key limitations of prior work like LoRA-Pro.
2. The experiments are comprehensive, covering diverse tasks and benchmarks. The results consistently show improvements over baselines, including LoRA-Pro, and the ablation studies add depth to the evaluation.

- Weaknesses
1. The paper acknowledges that AltLoRA+ loses transformation invariance due to second-order momentum (Appendix D.2). This is a significant limitation, yet the discussion is brief.
2. While AltLoRA narrows the gap, it doesn’t consistently match full fine-tuning (e.g., Table 2 shows Full FT outperforms AltLoRA on HumanEval). The claim of "narrowing the gap" could be tempered.

---

> ### Author Rebuttal · Authors · 2025-07-31
>
> We thank the reviewer for the insightful and constructive comments. Below, we provide our detailed responses to each point. We hope these clarifications help address your concerns.
>
> ---
> **W1: The discussion about AltLoRA+ and transformation invariance**
>
> We thank the reviewer for the comment. ***AltLoRA+ aims for accelerating AltLoRA instead of achieving transformation invariance***. That said, since our primary goal of AltLoRA+ is not to develop a fully transformation-invariant optimizer, we believe this limitation does not significantly impact its main contributions. In AltLoRA, our core contribution lies in inducing optimally aligned gradients and momentum within low-rank subspaces, which already demonstrates better performance compared to baseline methods. Besides the algorithm design, our theoretical guarantees, including stable feature learning, transformation invariance, and convergence analysis, provide a deeper understanding of the theoretical significance of gradient alignment in optimizing LoRA. To further ***accelerate convergence***, we propose AltLoRA+, which incorporates element-wise second-order momentum in a manner similar to AdamW (for the explanation of why element-wise second-order momentum will break transformation invariance, please refer to the answer of Q1). This enhancement leads to additional performance gains over the baselines. The improvement from AltLoRA to AltLoRA+ can also be regarded as an ablation study showcasing the benefits of momentum acceleration, which highlights the significance of AltLoRA+.
>
> ---
> **W2: Narrowing the gap**
>
> Our work aims to better understand the limitations of optimization within low-rank subspaces and to design more efficient algorithms that align low-rank updates closely with full-gradient directions. While we do not claim to completely close this gap, our empirical results demonstrate that our method brings low-rank adaptation closer to full fine-tuning in empirical performance. Following the reviewer's suggestion, we hope to revise our claim like "it narrows this gap toward full fine-tuning in two out of three natural language generation tasks and in the majority of natural language understanding tasks."
>
> ---
> **Q1: Analysis on transformation invariance and aligned second-order momentum**
>
>
> First, we will clarify why the second-order momentum in AltLoRA+ breaks transformation invariance. Here, we will prove AltLoRA+ breaks *scalar scale invariance* (See Def 2 in [1] for the formal definition), as a weaker version of transformation invariance. For two pairs of low-rank adaptation $ B_1 A_1 = B_2 A_2$, if we assume $A_2 = s A_1$, $B_2 = \frac{1}{s} B_1$, then
>
> $$\delta A_1 = -\eta \frac{\widetilde{\nabla }_{A_1} L}{\left(\nabla\_{A_1}L \odot \nabla\_{A_1}L\right)^{1/2}} = -\eta \frac{(B_1^\top B_1)^{-1} B_1^\top \nabla_w L}{\left(B_1^\top \nabla_w L \odot B_1^\top \nabla_w L\right)^{1/2}} = -\eta \frac{1}{s^2} \cdot \frac{(B_2^\top B_2)^{-1} B_2^\top \nabla_w L}{\left(B_2^\top \nabla_w L \odot B_2^\top \nabla_w L\right)^{1/2}} = \frac{1}{s^2} \delta A_2,$$
>
>
> where $\widetilde{\nabla }_{A_1} L$ is the scaled gradient. Therefore, the update would violate scalar scale invariance as $$B_1 \delta A_1 = s B_2 \cdot \frac{1}{s^2} \delta A_2 = \frac{1}{s} B_2 \delta A_2 \neq B_2 \delta A_2.$$
>
>
>
>
> Secondly, it is possible to align the second-order momentum with the full gradient in a principled way, but doing so does not necessarily improve performance. Let $V_t^B$ denote the second-order momentum of $B$ at time $t$. For example, analogous to aligning first-order momentum in the low-rank subspace, we can formulate a similar problem for second-order momentum as
>
> $$\min_{\widetilde{V}^B_t} \|\|(A_t)^T V_t^B A_t - (A_{t+1})^T \widetilde{V}^B_t A_{t+1} \|\|^2,$$
>
> where we assume $V_t^B$ is aligned to the low-rank space at time $t$. Here, the optimal solution is
>
> $$\widetilde{V}^B_t = (P_{A_{t+1}})^T V_t^B P_{A_{t+1}},$$
>
>  where $P_{A_{t+1}} = A_t A_{t+1}^T \left( A_{t+1} A_{t+1}^T \right)^{-1}.
> $ Then the updated second-order momentum for $B$ at time $(t+1)$ is $V\_{t+1}^B = \widetilde{V}^B_t + (\widetilde{\nabla}\_{B_t}L)^T\widetilde{\nabla}_{B_t}L$. However, our numerical study shows that the accuracy of this approach does not lead to performance gains. With the same setup in Table 2, the accuracy on the GSM8K test set is only 56.12. Thus, while principled alignment is possible, it appears less effective in practice. In contrast, AltLoRA+, which focuses on aligning first-order momentum efficiently, demonstrates stronger empirical performance.
>
>
>
> ---
> Thank you for your constructive feedback. We will incorporate the new results in the revised version, and we hope our responses satisfactorily address your concerns.
>
> ---
>
>
> [1] Yen J N, Si S, Meng Z, et al. LoRA Done RITE: Robust Invariant Transformation Equilibration for LoRA Optimization. arXiv preprint arXiv:2410.20625, 2024.

---

> > ### Comment · Reviewer_uqXj · 2025-08-03
> >
> > Thanks for the response. My score remains the same.

---

### Official Review · Reviewer_rKDu · 2025-06-30

**Clarity:** 2
**Significance:** 3
**Originality:** 3
**Rating:** 4
**Confidence:** 4

**Summary:**

This paper proposes AltLoRA which alternatively updates LoRA's two trainable components to simulate full matrix updates. Specifically, while fixing matrix $B$ and training matrix $A$, the authors consider an optimization problem that finds a proper subspace gradient $\nabla_A$ so that $sB\nabla_A$ has the smallest error to estimate $\nabla_W$. AltLoRA directly provides the closed-form solution for the problem.  Based on it, the authors derived the update strategy for the momentum matrix. The overall AltLoRA algorithm works better than other PEFT baselines in the experiments, enjoying a comparable or better training performance than full fine-tuning. Besides, this paper provides analysis on stable feature learning, transformation invariance, and convergence analysis on a two-layer ReLU NN tuning problem.

**Questions:**

i) I wonder whether the models are fine-tuned on specific tasks for only one epoch or more than one epochs. I'm asking because I did not find the number of epochs the models are trained on, which I believe is highly correlated with the meaning of accuracy results. Maybe I have missed it, but it should be included in the experimental details.

ii) Is it a common choice to use LLaMA3-8B-Base as a pretrained model in the literature? I'm asking because in my experience, LLaMA3-8B-Instruct can be fine-tuned to perform much better using vanilla LoRA, and results based on the instructive model should better reveals the practical fine-tuning ability.

iii) In Line 611, it is stated that "Random projection is also applied in Ga-LoRA([88])", however, the method proposed in [88] uses a deterministic projection strategy based on top-K + SVD, and its name should be "GaLore" instead of "Ga-LoRA". In the literature, a variant of GaLore using random projection is called GoLore, which is proposed in [arXiv:2410.11289].

**Ethical Concerns:**

["NO or VERY MINOR ethics concerns only"]

**Final Justification:**

Since most of my concerns are addressed, I keep my score as is.

**Limitations:**

Yes.

**Quality:**

3

**Strengths And Weaknesses:**

Strengths:

i) The proposed method appear noval and makes sense for me. The underlying intuition is quite natural and the presentation is easy to follow.

ii) The empirical results are promising, which I believe can address the potential of the proposed method.

iii) The theoretical result of transformation invariance makes the method more appealing to me.

Weaknesses:

i) The convergence analysis is relatively simple. It only considers a toy model and deterministic gradients. Noting that PEFT is for large-scale training tasks, it is better to consider stochastic gradients in both the algorithmic design and the analysis.

ii) The performance-efficiency trade-off is not clear. In my understanding, AltLoRA achieves better convergence result while being less memory- and computation-efficient than LoRA. It is recommended to present a trade-off curve regarding both perfomance and efficiency, which I believe can help better address the effectiveness of AltLoRA.

iii) To better illustate the effectiveness of AltLoRA, I expect to see training curves regarding the training or validation loss. However, the experimental results only include accuracy tables, leaving it unclear whether AltLoRA benefits from faster and more accurate convergence or it benefits from better generalization.

---

> ### Author Rebuttal · Authors · 2025-07-31
>
> We thank the reviewer for the insightful and constructive comments. Below, we provide our detailed responses to each point. We hope these clarifications help address your concerns.
>
> ---
> **W1: The convergence analysis and stochastic optimization**
>
> Our key contribution is to efficiently align both the gradient and momentum with the full gradient. The theoretical guarantees we establish provide insights into why the alignment intuition helps to improve performance in numerical studies. In contrast to the baseline work *LoRA-Pro*, which only proposed the theoretical motivation, our work provides formal guarantees on *stable feature learning, transformation invariance, and convergence*, thereby deepening the understanding of the importance of gradient alignment.
>
> In particular, for the convergence analysis, [1][2] also analyze LoRA on an over-parameterized two-layer ReLU neural network tuning problem. Our convergence analysis demonstrates that the proposed *alternating scaled gradient descent* method for LoRA achieves the same convergence rate as joint optimization (as established in [1] using the same toy model). More importantly, we show that the convergence rate of alternating scaled gradient descent is ***independent of the condition number***, and that alternating updates allow for a ***wider range of learning rates*** compared to joint updates. These results constitute the core theoretical contributions of our convergence analysis for alternating projection in LoRA training.
>
> Regarding the stochastic gradient, thank you for raising this point.
>
> First, *for the algorithm design*, we do implement stochastic batch gradients in our fine-tuning process, as it is standard practice in PEFT. In the revised version, we will add the corresponding description to clarify this in our Algorithms 1 and 2. For example, for the update of $A$ in ***L167***, we will  explicitly note: "Only backpropagate with respect to $A_t$ and obtain the stochastic batch gradient $\nabla_{A_t}L$."
>
> Second, while prior gradient alignment works also adopt stochastic gradients in practice, they typically do not analyze how stochastic gradients impacts the gradient alignment effect. Extending our theoretical analysis to explicitly incorporate stochastic gradient behavior is a promising direction for future work, particularly to better bridge the gap between LoRA and full fine-tuning from a stochastic optimization perspective.
>
> ---
> **W2: The performance and efficiency trade-off**
>
> Thank you for the helpful suggestion regarding visualization. We have reported the performance in Table 2 and the memory cost in Table 3. To better illustrate the memory-performance trade-off of our algorithm, we would like to include a scatter plot in the revised version of the paper. For each task, the x-axis will represent memory cost, while the y-axis will correspond to the performance metric (e.g., accuracy). This figure will visually demonstrate how our method improves over baselines. Here, we provide a table to describe the results for the GSM8K task by fine-tuning the Llama3.1-8B model, which supports that our algorithms achieve better performance than LoRA with similar memory efficiency.
>
>
> | Method                      | Memory Cost(GB) | Accuracy |
> |----------------------------|:-----:|:---------:|
> | Full FT                   | > 48  | 73.31 |
> | LoRA                      | 22.26  | 66.11 |
> | LoRA-Rite                 | 25.39 | 74.10 |
> | LoRA-Pro                  | 40.12 | 73.12 |
> | AltLoRA                   | 22.56 | 74.49 |
> | AltLoRA+                  | 23.16 | 76.91 |
>
> ---
> **W3: Training Curves**
>
>
>
> We provide the training loss in the table below. As shown, our method not only converges to a lower training loss than full fine-tuning, but also does so more rapidly than most baselines.
>
> The *accuracy metrics* reported in Table 2 are evaluated on the corresponding test sets, as described in Appendix E.1. Our method achieves higher scores, highlighting its superior generalization performance.
>
>
> | Method                     | step 500 | step 1000 | step 1500 | step 2000 | step 2500 | step 3000 |
> |----------------------------|:--------:|:--------:|:--------:|:--------:|:--------:|:--------:|
> | Full Fine-tuning           |     0.51     |     0.28     |    0.24      |    0.18      |    0.18     |     0.18     |
> | LoRA                   |     0.57     |     0.30     |     0.26     |     0.23     |   0.22      |   0.22       |
> | LoRA-Pro              |    0.51      |     0.31     |     0.24     |    0.22      |   0.21       |    0.21      |
> | LoRA-GA             |     0.45     |   0.24       |    0.20      |     0.18    |   0.18       |     0.18     |
> | LoRA-Rite             |     0.55     |    0.31      |    0.24      |    0.22      |     0.21     |     0.21     |
> | AltLoRA                 |     0.48     |     0.25     |     0.21     |    0.19      |   0.18      |       0.18   |
> | AltLoRA+                |    0.45      |    0.23      |     0.19     |     0.18     |    0.17      |    0.17      |
>
>
> ---
> **Q1: Epoch number**
>
>
> We have reported the number of training epochs in the experimental setup to avoid confusion. As reported in ***Line 897***, the iteration number was set to 1, which corresponds to a single training epoch.
>
>
> We agree that the number of epochs is an important training design, especially given its potential correlation with accuracy metrics. To further support the validity of our experiment, we add an ablation study for the training epoch with ***5 training epochs*** under the setup of Table 2. With the increase in training epochs, our approach would outperform baselines as well.
>
> | Method                     | MT‑Bench | GSM8K | HumanEval |
> |----------------------------|:--------:|:-----:|:---------:|
> | Full FT                        | 6.24±0.03 | 75.31±0.47 | 50.42±1.20 |
> | LoRA                           | 6.19±0.02 | 66.78±1.14 | 41.54±1.01 |
> | PiSSA                          | 6.11±0.03 | 67.39±0.61 | 42.12±1.45 |
> | rsLoRA                         | 6.18±0.02 | 67.98±1.21 | 44.31±1.31 |
> | LoRA+                          | 6.21±0.02 | 74.00±0.61 | 44.32±1.33 |
> | DoRA                           | 6.23±0.03 | 67.12±0.84 | 45.12±1.32 |
> | AdaLoRA                        | 6.05±0.02 | 68.29±0.98 | 42.58±1.84 |
> | LoRA‑GA                        | 6.24±0.05 | 73.44±1.10 | 45.56±1.48 |
> | LoRA‑Pro                       | 6.22±0.02 | 73.24±0.79 | 43.21±1.50 |
> | LoRA‑Rite                      | 6.20±0.03 | 73.48±0.65 | 45.46±1.21 |
> | **AltLoRA**                   | 6.26±0.04 | 74.52±0.37 | 46.28±1.11 |
> | **AltLoRA+**                   | 6.18±0.04 | 76.01±0.34 | 50.01±1.01 |
>
> ---
> **Q2: Llama3-8B-Base**
>
> Using the ***Llama3-8B-Base*** model is a common practice in the literature. Among the most relevant baselines, **LoRA-Pro** and **LoRA-GA** adopt either *Llama2-7B* or *Llama3-8B-Base* as the pretrained model. Furthermore, several recently published works [4][5][6] on LoRA also use *Llama3-8B-Base* as the standard backbone.
>
> For the *Llama3-8B-Instruct* model, we fine-tune it on the MetaMathQA dataset and evaluate it on the GSM8K test set, reporting accuracy as the primary metric. Our results demonstrate that even when using a stronger pretrained model, low-rank adaptation can still yield performance gains. More importantly, both AltLoRA and AltLoRA+ consistently outperform the standard LoRA baseline. We will add the full comparison in the revised revision.
>
>
> | Method                     | GSM8K |
> |----------------------------|:-----:|
> |Llama3-8B-instruct | 61.24 |
> | + Full FT           | 83.26 |
> | + LoRA              | 81.12 |
> | + AltLoRA           | 82.60 |
> | + AltLoRA+          | 83.88 |
>
>
> ---
> **Q3: Random projection and GaLore**
>
> We appreciate the clarification regarding the naming: it should be “GaLore“[5] instead of ”Ga-LoRA“. Additionally, we acknowledge that the variant using random projection is introduced in a separate work, ***GoLore***. And we will include GoLore into our related work as well. We will revise the text accordingly to correct both the description and the naming to avoid confusion.
>
> ---
> Thank you for your constructive feedback. We will incorporate the new results in the revised version, and we hope our responses satisfactorily address your concerns.
>
>
>
> ---
>
> [1] Zhang, F. and Pilanci, M. Riemannian Preconditioned LoRA for Fine-Tuning Foundation Models.  In International Conference on Machine Learning (pp. 59641-59669). PMLR.
>
> [2] He, Y., Li, P., Hu, Y., Chen, C., and Yuan, K. Subspace Optimization for Large Language Models with Convergence Guarantees. In Forty-second International Conference on Machine Learning. 2025.
>
> [3] Zhang, J., Chiu, H.M., and Zhang, R.Y. Accelerating SGD for Highly Ill-Conditioned Huge-Scale Online Matrix Completion. Advances in Neural Information Processing Systems 35 (2022): 37549-37562.
>
> [4] Liao, X., Li, S., Xu, Y., Li, Z., Liu, Y. and He, Y. GaLore+: Boosting Low-Rank Adaptation for LLMs with Cross-Head Projection. arXiv e-prints, pp.arXiv-2412.
>
> [5] Zhang, C., Lianhai, R.E.N., Cheng, J. and Li, Q. From Weight-Based to State-Based Fine-Tuning: Further Memory Reduction on LoRA with Parallel Control. In Forty-second International Conference on Machine Learning. 2025.
>
> [6] Huang, Q., Ko, T., Zhuang, Z., Tang, L., and Zhang, Y. HiRA: Parameter-Efficient Hadamard High-Rank Adaptation for Large Language Models. In The Thirteenth International Conference on Learning Representations. 2025.
>
>
> [7] Recht, B., Maryam F., and Parrilo, P. Guaranteed minimum-rank solutions of linear matrix equations via nuclear norm minimization. SIAM Review 52.3 (2010): 471-501.

---

> > ### Comment · Reviewer_rKDu · 2025-08-01
> >
> > Thanks for the detailed rebuttal. Most of my concerns have been well addressed, and I have no further questions.

---

### Official Review · Reviewer_4vyq · 2025-07-02

**Clarity:** 3
**Significance:** 3
**Originality:** 3
**Rating:** 5
**Confidence:** 3

**Summary:**

This paper proposes AltLoRA, a new PEFT method that narrows the performance gap between LoRA and full fine-tuning.

The key idea is to update the low-rank matrices $A$ and $B$ via alternating projections onto their respective subspaces, with each step having a closed-form solution. A momentum mechanism is also introduced, aligned to the low-rank space, avoiding the memory overhead of full-size momentum.

The authors provide theoretical analysis (stability, invariance, convergence) and show strong empirical results across language tasks. Both AltLoRA and its variant AltLoRA+ outperform prior LoRA methods and approach full fine-tuning performance, while retaining LoRA’s efficiency.

**Questions:**

See weakness.

**Ethical Concerns:**

["NO or VERY MINOR ethics concerns only"]

**Final Justification:**

The author resolved my concern during the rebuttal. I have decided to keep my score unchanged and recommend acceptance.

**Limitations:**

yes

**Paper Formatting Concerns:**

According to the guideline, all tables must be centered, neat, clean and legible. The table number and title always appear before the table. But the table 2 does not follow this.

**Quality:**

4

**Strengths And Weaknesses:**

Strength
1. Novolty: The alternating update scheme for low-rank matrices is an elegant and original solution to the gradient approximation problem. It avoids issues with non-uniqueness in joint methods and naturally leads to a principled momentum alignment strategy that is both effective and memory-efficient.
2. Soundness: The paper offers strong theoretical guarantees and extensive experiments across models (Llama-3, T5) and tasks (math, code, dialogue, NLU) show that AltLoRA and AltLoRA+ consistently outperform prior LoRA methods and approach or exceed full fine-tuning, while retaining LoRA’s efficiency.

Weakness
1. Writing Quality: The paper has several presentation issues. For example, Table 2's caption is below the table, unlike the others. Also, in line 601, the description of LoRA-Pro ([66]) is inaccurate—its main contribution is modifying gradients during training, not just initialization ([65]). This is likely a typo, but it could confuse readers.
2. Missing Ablation on Momentum Alignment: While the momentum alignment mechanism is theoretically sound, the paper lacks an ablation comparing it with standard momentum updates. It would be helpful to know whether standard momentum leads to divergence or if it performs similarly, to better assess the necessity of the alignment.


Overall, I think it's a good paper if the authors can fix the weakness in the camera-ready version.

---

> ### Author Rebuttal · Authors · 2025-07-31
>
> The authors thank the reviewer for insightful and constructive comments. Below, we provide our detailed responses to each point. We hope these clarifications help address your concerns.
>
> ---
> **W1: Writing Quality**
>
> Thank you for bringing these presentation issues to our attention. We will relocate the caption of Table 2 to the correct position in the revised version. In ***L601***, the reference should be “***LoRA-GA***[65]” rather than “***LoRA-Pro***~[66]”. LoRA-GA indeed introduces a novel initialization approach that more closely aligns LoRA with full fine-tuning at the first step.
>
>
> ---
> **W2: Missing Ablation on Momentum Alignment**
>
> We include ablation studies to evaluate the effect of momentum alignment. The ***AltLoRA w/o MA*** variant differs from AltLoRA only in that momentum is not projected onto the low-rank subspace (momentum alignment). Across the tasks of natural language generation, AltLoRA outperforms AltLoRA w/o MA, achieving higher accuracy and lower variance. This demonstrates the advantages of aligning momentum with low-rank spaces.
>
>
>
> | Method                      | GSM8K | HumanEval |
> |----------------------------|:-----:|:---------:|
> | AltLoRA                       | 74.49±0.57 | 45.91±1.14 |
> | AltLoRA w/o MA                | 72.93±0.88 | 44.02±1.42 |
>
>
> ---
>
> Thank you for your constructive feedback. We will incorporate the new results in the revised version, and we hope our responses satisfactorily address your concerns.

---

> > ### Comment · Reviewer_4vyq · 2025-08-01
> >
> > Thank you for the detailed responses and clarifications. Based on your rebuttal, I have decided to keep my score unchanged and recommend acceptance.

---

### Official Review · Reviewer_tXEi · 2025-07-03

**Clarity:** 3
**Significance:** 3
**Originality:** 2
**Rating:** 4
**Confidence:** 4

**Summary:**

This paper propose a novel method called AltLoRA for aligning full finetuning gradients. It change the gradient update jointly on A and B (like lorapro does) to iteratively update A then update B respectively. Based on lorapro theory, they derive simpler solution then lorapro.  Numerical experiments demonstrate that AltLoRA outperforming vanilla LoRA and its variants on several benchmark datasets.

**Questions:**

See weakness

**Ethical Concerns:**

["NO or VERY MINOR ethics concerns only"]

**Final Justification:**

I raise my rating to a weak accept, in recognition of the extensive experiments and the interesting improvements that (aligned) alternating updates can bring. However, I maintain reservations regarding the flawed theoretical analysis and the imprecise descriptions in the paper.

**Limitations:**

The theoretical framework contains flaws (W1), and the optimization method appears unreliable (W3). The authors evaluate all baselines using only 1 epoch, which is insufficient for convergence for most baselines (W2).

**Quality:**

3

**Strengths And Weaknesses:**

### Strengths
- This paper provides a novel perspective on the initialization of LoRA from the gradient updation aspect.
- The paper is theoretically comprehensive, well-written, and easy to follow.
- The method demonstrates promising performance and exhibits robust behavior across different hyperparameter settings, particularly for the sensitive parameters like Alpha and learning rate in LoRA.

### Weaknesses
- W1. The theoretical framework contains flaws: as established, B is zero-initialized in LoRA, but in Algorithm 1 at $t=0$, updating A requires computing the inverse of $(B^TB)^{-1}$, which leads to undefined behavior. Furthermore, after several update steps when $t>0$, B contains minimal values (due to small learning rates), causing $(B^TB)^{-1}$ to become extremely large. This consequently makes the gradients for A excessively large and corrupts the weight of A. Since we typically use fp16/bf16 mixed-precision training in practice, these large backpropagated updates may exceed numerical ranges. These two factors may cause the loss to diverge and potentially lead to training collapse.
- W2. The authors evaluate all baselines using only 1 epoch (L891), making convergence speed the primary determinant of reported performance. This evaluation protocol may disadvantage LoRA-based methods, as faster initial convergence does not necessarily indicate better final performance, slower-converging methods may ultimately reach superior solutions. For reliable comparisons, experiments should run for at least 5 epochs (or more for hard-to-converge tasks) to obtain reliable performance measurements.
- W3. While standard LoRA jointly optimizes both variables A and B (M1), AltLoRA decouples this into sequential optimization - first A, then B (M2). However, the convergence properties for LoRA with M1 optimization remain unproven (D3 only discusses a 2-layer ReLU NN case without providing proofs for general LoRA scenarios. Furthermore, practical training typically involves more complex neural architectures). Empirical evidence suggests that alternately freezing layers during training often leads to training instability rather than performance improvement. Furthermore, since M2 is essentially a degenerate case of M1 with stricter convexity requirements, the local minima found by M2 should theoretically be inferior to those found by M1. Therefore, the observed superior performance of M2 over M1 in Table 2 and Table 4 presents an unexplained anomaly that warrants further investigation.
- W4. In Table 1, the performance seems degrade as rank increasing, which is strange.
- W5. Given the theoretical concerns raised in W1, AltLoRA’s superior performance over all baselines appears unexpected. Table 8's finding that zero initialization of B outperforms Gaussian initialization contradicts the issue identified in W1. I personally consider it may related with W2. Due to these issues,  I currently maintain reservations about the methodological validity and choose to assign a low rating temporarily.  Upon receiving adequate explanation of these points, I am happy to revisit my assessment and raising the scores.

---

> ### Author Rebuttal · Authors · 2025-07-31
>
> The authors thank the reviewer for insightful and constructive comments. Below, we provide our detailed responses to each point. We hope these clarifications help address your concerns.
> ---
> **W1: Computing $(B^TB)^{-1}$**
>
> Sorry for the confusion. Please note that we have clarified at ***L121*** and ***L149*** of the original submission that we didn't directly compute $(B^\top B)^{-1}$ but used a regularized inverse instead. We use $(B_t^\top B_t)^{-1}$ in Algorithms 1 and 2 just for notational simplicity.
>
> To be more specific, the full-rank assumption for $B$ is typically not satisfied during training according to our analysis. Thus, following the reasoning in our gradient-alignment analysis (see *Corollaries 1 and 2*), we introduce weight decay during the gradient alignment and compute the well-defined regularized inverse:
> $ (B^\top B + \lambda \mathbb{I})^{-1}.$
> This formulation ensures numerical stability and avoids undefined behavior during updates. In all experiments of the original submission, we have used $\lambda=1e^{-5}$.
>
> We will revise the algorithm descriptions to replace $(B_t^\top B_t)^{-1}$ with $(B_t^\top B_t + \lambda \mathbb{I})^{-1}$ to avoid any ambiguity.
>
> ---
> **W2: Increasing training epoch**
>
>
> Thank you for the helpful suggestion. For fairness and direct comparability, our original experiments followed the evaluation protocol used in *LoRA-GA* (and *LoRA-Pro*), which trains for ***one*** epoch, and we reported the corresponding results.
>
> To provide a more comprehensive evaluation, as the reviewer suggested, we have ***re-run all experiments in Table 2 using five epochs***. The full results are reported below. We observe that our method continues to consistently outperform all baselines under this extended training regime, demonstrating that the improvements are robust beyond early convergence and hold across longer fine-tuning schedules. We would be glad to include these results in the revised manuscript for completeness.
>
>
>
> | Method                     | MT‑Bench | GSM8K | HumanEval |
> |----------------------------|:--------:|:-----:|:---------:|
> | Full FT                        | 6.24±0.03 | 75.31±0.47 | 50.42±1.20 |
> | LoRA                           | 6.19±0.02 | 66.78±1.14 | 41.54±1.01 |
> | PiSSA                          | 6.11±0.03 | 67.39±0.61 | 42.12±1.45 |
> | rsLoRA                         | 6.18±0.02 | 67.98±1.21 | 44.31±1.31 |
> | LoRA+                          | 6.21±0.02 | 74.00±0.61 | 44.32±1.33 |
> | DoRA                           | 6.23±0.03 | 67.12±0.84 | 45.12±1.32 |
> | AdaLoRA                        | 6.05±0.02 | 68.29±0.98 | 42.58±1.84 |
> | LoRA‑GA                        | 6.24±0.05 | 73.44±1.10 | 45.56±1.48 |
> | LoRA‑Pro                       | 6.22±0.02 | 73.24±0.79 | 43.21±1.50 |
> | LoRA‑Rite                      | 6.20±0.03 | 73.48±0.65 | 45.46±1.21 |
> | **AltLoRA**                   | 6.26±0.04 | 74.52±0.37 | 46.28±1.11 |
> | **AltLoRA+**                   | 6.18±0.04 | 76.01±0.34 | 50.01±1.01 |
>
> ---
> **W3: Why AltLoRA outperforms standard LoRA**
>
> We respectfully disagree with some of the reviewer's opinions and aim to clarify the reasons behind the improved performance of our method.
>
> First and foremost, unlike full fine-tuning, the gradients in standard LoRA are confined to low-rank subspaces, which limits its effectiveness. Prior work, such as LoRA-Pro, has identified this limitation and attempted to align LoRA gradients to approximate the full gradient. However, LoRA-Pro does not yield a unique update solution and demonstrates unstable performance. In contrast, our alternative optimization strategy addresses this issue by ***better aligning the low-rank gradient with the true full gradient***. This alignment underlies the superior performance of AltLoRA compared to standard LoRA.
>
> Regarding theoretical analysis, we emphasize that our work is not purely theoretical or solely focused on convergence proofs. Instead, our goal is to develop an effective LoRA algorithm ***supported by theoretical insights*** that justify our design choices. Specifically, we have demonstrated that our approach achieves stable feature learning and transformation invariance—property not present in standard LoRA but crucial for non-asymptotic training behavior. Furthermore, the convergence guarantee, albeit established on a simplified two-layer ReLU network, provides additional evidence of the method’s design soundness.
>
> Finally, we note that the strategy of freezing layers, which may appear as an *empirical heuristic*, fundamentally differs from our alternating optimization approach. Alternating optimization is a well-established technique with decades of study. Recent works—[1] in LoRA and [2] in matrix factorization—also show that alternating optimization can be more stable and effective than joint optimization in certain settings.
>
> ---
> **W4: Performance as rank increases**
>
> In Table 2, we observe that performance degrades as the rank increases. A similar trend appears in the original LoRA paper (Tables 15 and 18 in [3]). We hypothesize that larger ranks introducing more trainable degrees of freedom can easily lead to overfitting when learning from a small sample size. Consistent with this view, in the five-epoch setting, the performance of AltLoRA and AltLoRA+ with higher rank improves and exceeds the default setting (rank = 8).
>
>
> | Method                     | MT‑Bench | GSM8K | HumanEval |
> |----------------------------|:--------:|:-----:|:---------:|
> | **AltLoRA**                   | 6.26±0.04 | 74.52±0.37 | 45.41±1.11 |
> | **AltLoRA** (rank=32)    | 6.13±0.03  | 74.92±0.67 | 46.82±1.34 |
> | **AltLoRA** (rank=128)      | 6.28±0.04  | 74.71±0.56 | 46.88±1.38 |
> | **AltLoRA+**                   | 6.18±0.04 | 76.01±0.34 | 50.01±1.01 |
> | **AltLoRA+** (rank=32)         | 6.18±0.05 | 77.69±0.41 | 50.21±1.43 |
> | **AltLoRA+** (rank=128)        | 6.14±0.04 | 77.21±0.67 | 50.44±1.22 |
>
> ---
> **W5: The superiority of AltLoRA**
>
>
> By computing $(B^{\top}B + \lambda\mathbb{I})^{-1} $ instead of $(B^{\top}B)^{-1} $, we eliminate the theoretical concern raised in *W1*. As a result, the downstream argument in *W5*, which relies on *W1*, no longer applies. To examine whether the superior performance of zero initialization over Gaussian initialization for $B$ is related to *W2*, we revisit the same setup in Table~8 under a 5-epoch training schedule. We find that, even with more steps, zero initialization of $B$ yields a higher accuracy (74.52) compared to Gaussian initialization (74.31), which shows a consistent result with our manuscript.
>
> ---
> Thank you for your constructive feedback. We will incorporate the new results in the revised version, and we hope our responses satisfactorily address your concerns.
>
>
> ---
> [1] Chen S, Guo Y, Ju Y, et al. Robust federated finetuning of llms via alternating optimization of lora. arXiv preprint arXiv:2502.01755, 2025.
>
>
> [2] Jia, Xixi, et al. Preconditioning matters: Fast global convergence of non-convex matrix factorization via scaled gradient descent. Advances in Neural Information Processing Systems 36 (2023): 76202-76213.
>
> [3] Hu E J, Shen Y, Wallis P, et al. Lora: Low-rank adaptation of large language models. International Conference on Learning Representations, 2022.

---

> > ### Comment · Reviewer_tXEi · 2025-08-04
> >
> > I appreciate the authors' kind response! My concerns regard insufficient epochs and ranks are addressed. However:
> >
> > -  I still not convinced in computing $(B^{\top}B + \lambda\mathbb{I})^{-1} $ instead of $(B^{\top}B)^{-1} $:
> >     -   In L119, you mention that adding $\lambda \mathbb {I} $ ensures $B_t$ and $A_t$ are full-rank. However, this is unrelated to the concern I originally raised. My concern is not about rank deficiency, but about gradient explosion when $B_t^\top B_t$ becomes very small (especially during early training). In that case, the proposed gradient update: $\frac{1}{s^2}(B_t^\top B_t + \lambda \mathbb {I} )^{-1} \nabla_{A_t} L$ will still approximate $\frac{1}{s^2 \lambda} \nabla_{A_t} L$ when $B_t^\top B_t \to 0$. Given $\lambda = 1e^{-5}$, this leads to a large gradient, thus failing to address the instability you claim to fix.
> >     -   Your main theorems (e.g., Theorem 1 and Eq. (9)-(13)) are derived under the exact closed-form update given in Equation (8), which has no regularization. However, your rebuttal claims you actually use the regularized version derived from a different optimization problem (Eq. (18)). These two problems yield different closed-form solutions. As a result, I don't think the new gradient fulfills the inital objective (Equation (8)) and it's unclear whether transformation invariance or the convergence guarantees established under Eq. (8) still hold for the actual implementation using Eq. (18).
> >     -   You mention that the $\lambda \mathbb {I} $  is like "weight decay" (L119). However, the regularized $ (B_t^T B_t + \lambda\mathbb{I} ) $ is actually the solution to L2 regluration. A true weight decay update would take the form $ \frac{1}{s^2} (B_t^T B_t )^{-1} \nabla_{A_t} L+ \lambda |A_t| $ which is conceptually and mathematically distinct. I believe the current explanation is imprecise and misleading.
> > -   On the Sequential (Alternating) vs. Joint Optimization of A and B:
> >     -   My concern is not about why AltLoRA outperforms LoRA, but why sequentially optimizing $A$ and $B$ (M2) is superior to joint optimization (M1). Since M2 is effectively optimizing over a restricted subspace of the joint space of M1, it should not explore better optima in theory. Thus, unless strongly justified, M2 should not be inherently better than M1. As the alternating update is the core insight in this paper (the alignment is alreay explored in LoRA-Pro), I think it worth in-depth discussion.
> >     -  While the analysis in D3.2 ensures convergence of M2, it does not imply that M2 outperforms M1.
> >     -  Your claim that alternating updates are "well-established" contradicts practice in recent PEFT literature. Prominent methods such as DoRA, PiSSA, rsLoRA, LoRA-GA, and LoRA-Pro all adopt joint updates. Even in full model fine-tuning, alternating updates are rare. Therefore, claiming M2 as a better strategy requires stronger theoretical justification, which is currently lacking in the paper.
> >     -  Could you please specify the update form for the "joint update" in Table 6? For the "separate update", the gradient is derived from Equation (8), but it is unclear what the corresponding update rule is for the "joint update". This lack of clarity makes it difficult to compare the two approaches.
> >
> > As a result, I feel that the theoretical analysis and experimental results are not well integrated.

---

> ### Author Response · Authors · 2025-08-07
>
> Thank you for your kind response and for acknowledging our clarifications regarding epochs and ranks. Below, we further address and clarify the remaining concerns.
>
> ---
>
> $\bullet$  **On computing $(B^TB + \lambda \mathbb{I})^{-1}$**
>
> $\circ$ We now understand the reviewer’s original concern is not about rank deficiency but rather gradient instability when $B_t^TB_t\approx 0$, which is a relevant issue in early training. We appreciate the clarification. In our earlier response, we mentioned rank deficiency because the reviewer referenced the “undefined behavior” of $(B^TB)^{-1}$, which is typically ill-posed when $B_t$ is not of full rank.
>
> **Regarding the reviewer's concern about gradient explosion**, we observe empirically that the update is stable even when $B_t^TB_t$ is small in early training steps. This stability is primarily due to the following algorithm design: we follow LoRA-Pro and adopt a cosine learning rate scheduler with a **warmup** ratio of 0.03. During the warm-up period (when $B_t^TB_t$ may be small), the magnitude of the learning rate starts at 0 and gradually increases to the default value (i.e., grid search $\\{1e^{-4},4e^{-5},1e^{-5} \\}$ in AltLoRA+). As a result, even when $B_t^TB_t$ is small, the effective step size for updating $A$ is controlled and does not lead to exploding gradient updates. Specifically, the magnitude of the update during early steps is approximately $\frac{\eta}{s^2\lambda}\nabla_{A_t}L$,
> which remains well-behaved due to small $\eta$ during warm-up. We support this with empirical evidence below, showing the norms of $A$ and $B$ updates as well as the loss remain stable during the early steps.
>
>
> For AltLoRA:
> | Metric | step 1 | step 20 | step 40 | step 60 | step 80 | step 100 |
> |-----|:--------:|:--------:|:--------:|:--------:|:--------:|:--------:|
> | $\|A\|$ | 0.0039 | 0.0011  | 0.0014 | 0.0011 |0.0009 | 0.0010 |
> | $\|B\|$ | < 1e-4 |< 1e-4|< 1e-4 |< 1e-4 |< 1e-4| < 1e-4 |
> | Loss | 1.0111 | 0.9302 | 0.7744 | 0.6881 | 0.6921 | 0.7058 |
>
>
> For AltLoRA+:
> | Metric  | step 1 | step 20 | step 40 | step 60 | step 80 | step 100 |
> |-----|:--------:|:--------:|:--------:|:--------:|:--------:|:--------:|
> | $\|A\|$ | 0.0022 |0.0010 | 0.0004|0.0004|0.0004|0.0004 |
> | $\|B\|$ | 0.0099 |0.0035| 0.0021 |0.0017 |0.0015| 0.0014 |
> | Loss | 1.0111 | 0.8411 | 0.6940 | 0.6433 | 0.6493 | 0.6770 |
>
> $\circ$  **Regarding the difference between Eq. (8) and Eq. (18)**, such a gap between the theoretical formulation and the practical implementation is *consistent with existing practices in the LoRA literature*. Prior works such as scaled AdamW [1] and LoRA-One [2] also precondition their gradients using $(B^\top B)^{-1}$ in theory, but apply $(B^\top B + \lambda \mathbb{I})^{-1}$ in practice for numerical stability. Notably, ***neither [1] nor [2] analyzes the regularized scaled gradient*** in their convergence guarantees. Our work closely follows this convention. Besides, LoRA-Pro assumes that the low-rank matrix is full-rank, yet also acknowledges that this assumption can be violated in early training  (shown in Table 2 of LoRA-Pro). For convergence analysis, LoRA-Pro didn't establish convergence analysis.
>
> Although the exact *transformation invariance* property of Eq. (8) may not strictly hold under Eq. (18), we find that in practice, the behavior of the update is very close due to the small value of $\lambda$. As we use a very small $\lambda$ (1e-5) in the implementation, a first-order Taylor expansion shows that $\frac{1}{s^2}(B_t^TB_t + \lambda\mathbb{I})^{-1}\nabla_{A_t}L \approx \frac{1}{s^2}(B_t^TB_t)^{-1}\nabla_{A_t}L$ up to the linear and higher order terms in $\lambda$, and in the special case where $\lambda = 0$, the two formulations are exactly equivalent. As such, the regularized update closely approximates the unregularized solution. For this reason, we conduct our theoretical analysis using the unregularized update in Eq. (8), which enables cleaner theoretical guarantees. Moreover, the stability and convergence observed in empirical results suggest that the effect of this perturbation is negligible in practice.
>
> Analyzing convergence for the scaled gradient with adding term $\lambda$ is significantly more challenging, since the hyperparameter $\lambda$ affects both the effective $L$-smoothness and the PL inequality in [3], and may require a customized spectral initialization as discussed in [4]. Understanding how $\lambda$ influences training dynamics and generalization in LoRA is an open question and a promising direction for future work.
>
>
> $\circ$ On "weight decay", we agree that our earlier phrasing may have been imprecise. To avoid this confusion, we will revise the sentence in L121 as follows:
>
> *"To mitigate this issue, we apply a Frobenius norm regularization to the two approximated gradients, which can stabilize the updates and remove the assumption of full rank."*

---

> ### Author Response · Authors · 2025-08-07
>
> $\bullet$  **On the alternating updates**
>
> $\circ$ We appreciate the reviewer’s thoughtful follow-up and would like to clarify that **we do not claim that alternating optimization (M2) is inherently better or universally superior to joint optimization (M1)**. Rather, our argument is contextual and specific to the gradient alignment setting, where joint optimization often fails to yield a unique solution and can be empirically unstable, as noted in LoRA-Pro (Appendix D.1). In contrast, alternating updates lead to unique closed-form solutions and stronger theoretical guarantees (stable feature learning and transformation invariance) over literature, which would help to explain the superity of our methods over baseline.
>
> **Regarding the concern that M2 operates in a restricted subspace of M1, we respectfully disagree with this characterization.** We note that truly restricted subspace optimization occurs in methods like FFA-LoRA [5], where only one sub-module (e.g., $A$) is optimized while the other ($B$) is frozen throughout training. In contrast, our alternating strategy updates both $A$ and $B$, sequentially projecting the full gradient onto the low-rank subspaces defined by $A$ and $B$, without freezing or excluding any parameter block.
>
> **Regarding the comparison with the alignment in LoRA-Pro and our core insight**, LoRA-Pro pioneered the idea of gradient alignment, and it does so via joint updates over two low-rank spaces $A$ and $B$, which can overlap, making it impossible to derive a unique optimal update, as acknowledged in their analysis. This overlapping leads to empirical instability (see Appendix D.1 in LoRA-Pro). In contrast, our work demonstrates that alternating updates, by projecting the full gradient sequentially, avoid this overlap and result in more stable and efficient updates. Specifically, in LoRA-Pro’s formulation, for different pairs of gradient $A$ and $B$, the full model update is the same: project to the low-rank spaces of $A$ and $B$ $$\frac{1}{s}B_t(B_t^TB_t)^{-1}B_t^T(\nabla_{W_t}L) + (\nabla_{W_t} L)A_t^T(A_tA_t^T)^{-1}A_t,$$ and compensate for the overlapping part $$-B_t(B_t^TB_t)^{-1}B_t^T(\nabla_{W_t}L)A_t^T(A_tA_t^T)^{-1}A_t.$$ In contrast, our alternating projection avoids such overlap by updating one submodule at a time. We further extend gradient alignment to momentum alignment, making the method more practical and optimizer-friendly within LoRA’s memory budget. In summary, our contribution is not a blanket preference for alternating updates, but rather a principled design that improves stability, performance, and theoretical tractability in the context of gradient alignment.
>
>
> $\circ$ Thank you for the observation. Our convergence analysis in Appendix D.3.2 establishes theoretical guarantees for the alternating update (M2), but does not claim it universally outperforms the joint update (M1). Our analysis demonstrates linear convergence at a rate independent of the condition number, which supports the theoretical soundness of the alternating scheme. However, *we do not claim that this implies M2 is superior to M1 in all settings.* We acknowledge that the original phrasing may have caused some confusion, and we will revise the text to make it clear that the theoretical guarantees are specific to the alternating scheme and are not intended as a universal comparison.
>
> $\circ$ Thank you for the helpful clarification. Our use of the term "well-established" refers to the *broader optimization literature*, where alternating updates are commonly used for problems involving parameterized low-rank matrix decompositions [6,7], due to their numerical stability and ease of implementation. While we acknowledge that joint updates are more prevalent in recent PEFT works, alternating strategies have also been adopted in some recent works, such as [8].
>
> Beyond PEFT, alternating updates are not rare in full-model fine-tuning. For instance, several recent approaches alternate between freezing and unfreezing subsets of layers during fine-tuning to improve stability and performance [9,10]. Furthermore, standard reinforcement learning settings routinely employ alternating updates, most notably between the actor and critic networks [11].
>
> Our manuscript explicitly discusses and compares methods that adopt joint updates, including those mentioned by the reviewer, and we provide fair empirical evaluations following the evaluation protocols of LoRA-Pro and LoRA-GA. We agree that a stronger theoretical justification for alternating updates in the PEFT context is valuable, and our current convergence analysis aims to be a step in that direction. We will be careful with the wording in the revised version to avoid overstating the prevalence of alternating updates in the PEFT literature.
>
>
> $\circ$ Thank you for pointing this out! The ``joint update" in Table 6 refers to *joint scaled gradient descent* with regularization. We will clarify this explicitly in the final version to avoid confusion.

---

> ### Author Response · Authors · 2025-08-07
>
> **Summary**
>
> We appreciate the reviewer’s comment. Our PEFT methods provide improved gradient alignment through **alternating projection** and introduce a novel **momentum alignment** strategy. These techniques are supported not only by theoretical guarantees, including **stable feature learning** and **transformation invariance**, but also by fair and consistent empirical comparisons following the protocols of prior work.
>
> Importantly, we have made concrete efforts to better integrate theory and practice. Our theoretical analysis focuses on the unregularized update, which is consistent with existing practices in the LoRA literature [1,2], and we adopt the regularized inverse $(B^TB+\lambda\mathbb{I})^{-1}$ during implementation, which is commonly used in practice for numerical stability [2].
>
> ---
>
>
>
>
>
>
>
> [1] Zhang F, Pilanci M. Riemannian preconditioned LoRA for fine-tuning foundation models. ICML, 2024.
>
> [2] Zhang Y, Liu F, Chen Y. LoRA-One: One-Step Full Gradient Could Suffice for Fine-Tuning Large Language Models, Provably and Efficiently. ICML, 2025.
>
>
> [3] Cheng C, Zhao Z. Accelerating gradient descent for over-parameterized asymmetric low-rank matrix sensing via preconditioning. ICASSP, 2024.
>
> [4] Xu X, Shen Y, Chi Y, Ma C. The power of preconditioning in overparameterized low-rank matrix sensing. ICML, 2023.
>
>
> [5] Sun Y, Li Z, Li Y, Ding B. Improving LoRA in privacy-preserving federated learning. ICLR, 2024.
>
>
> [6] Liu Z, Han Z, Tang Y, et al. Efficient over-parameterized matrix sensing from noisy measurements via alternating preconditioned gradient descent. arXiv:2502.00463.
>
> [7] Ward R, Kolda T. Convergence of alternating gradient descent for matrix factorization. NeurIPS, 2023.
>
>
> [8] Chen S, Guo Y, Ju Y, et al. Robust federated finetuning of llms via alternating optimization of LoRA. arXiv:2502.01755, 2025.
>
>
> [9] Pan R, Liu X, Diao S, et al. LISA: Layerwise importance sampling for memory-efficient large language model fine-tuning. NeurIPS, 2024.
>
> [10] Luo Q, Yu H, Li X. BAdam: A Memory Efficient Full Parameter Optimization Method for Large Language Models. NeurIPS, 2024.
>
> [11] Sheng G, Zhang C, Ye Z, et al. Hybridflow: A flexible and efficient RLHF framework. EuroSys, 2025.

---

> > ### Comment · Reviewer_tXEi · 2025-08-09
> >
> > Thank you for the detail response!
> >
> > Firstly, using a small $\lambda$ to mitigate the reverse issue, and further compensating the small $\lambda$ with a small learning rate during warmup, does not seem like an elegant solution. In fact, the use of small $\lambda$ is entirely excluded from the theoretical framework proposed in the paper. Given that this adjustment was not explicitly discussed in the paper or the first round of rebuttal, it appears to be more of a makeshift fix rather than a well-considered theoretical refinement. It is a bit overclaim that `These techniques are supported not only by theoretical guarantees, including stable feature learning and transformation invariance`.
> >
> > Secondly, for the alternating optimization (M2) and joint optimization (M1), I agree with the author's points that (1) alternating optimization in this paper is better than the sub-module optimization (FFA-LoRA); (2)  LoRA-Pro has some empirical instability (though, I think it may primarily comes from $(B^TB)^{-1}$ and zero initalization for $B$).  I thank the author for listing more alternating optimization series work but I need to emphasis that actor-critic methods are different from alternating optimization. Because in actor-critic methods both networks are updated in each iteration, the critic is usually updated more frequently. Alternating optimization, however, exclusively mask one out when update the other.  Though the $A$ and $B$ continously receive gradient, they cannot be updated together. The decoupled gradient flow transforms complex loss landscapes (with saddle points and local minima) into simplified mesh surfaces. This architectural bias fundamentally alters convergence properties, yet the paper’s discussion of its theoretical implications is imprecise and oversimplified. Given that convergence under alternating updates constitutes the core theoretical justification for the proposed method, this omission significantly undermines the soundness of the paper.
> >
> > As a result, I will raise my rating to a weak accept, in recognition of the extensive experiments and the interesting improvements that (aligned) alternating updates can bring. However, I maintain reservations regarding the flawed theoretical analysis and the imprecise descriptions in the paper.

---

> > > ### Author Response · Authors · 2025-08-09
> > >
> > > Thanks for recognizing our contribution!
> > >
> > > ---
> > >
> > > **Training design in $\lambda$ and learning rate**
> > >
> > > As stated in L650, we adopt the default value of $\lambda$ to preserve irreversibility, and as noted in L260, our choice of learning rate schedule follows the baseline LoRA-Pro. Therefore, our experimental design is not a "makeshift fix" for the early training stage. We would like to provide a corresponding analysis of warm-up stability in the manuscript.
> > >
> > > **Alternating Optimization**
> > >
> > > Thank you for your agreement. Regarding the instability of LoRA-Pro, we observe that it depends heavily on an auxiliary variable that significantly impacts performance. Table 6 in Appendix D.1 of LoRA-Pro, which reports results for different choices of this variable, supports our observation. For the convergence analysis, we follow the setup in [1] and establish linear convergence at a rate independent of the condition number, with greater tolerance to a wider range of learning rates. Our theoretical justification of our methods covers three key aspects (stable feature learning, transformation invariance and convergence guarantee), as highlighted in Section 4. We will revise the manuscript to make this description more precise.
> > >
> > > [1] Zhang F, Pilanci M. Riemannian preconditioned LoRA for fine-tuning foundation models. ICML, 2024.

---

> > > > ### Comment · Reviewer_tXEi · 2025-08-09
> > > >
> > > > Thank you for the further clarifications. I would like to respectfully emphasize the concern: while the authors note that the small $\lambda$ and the warmup procedure are part of the "default" setup, these components are not incorporated into the theoretical analysis based on Equation (8). This raises the question of whether the theory remains valid when these "default" choices, such as the specific $\lambda$ , the use of warmup, or the particular warmup strategy, are altered or omitted. In this sense, the current theoretical framework appears incomplete, and these practical fixes seem more like ad hoc fix rather than well-integrated refinements. I believe the description could be clarified to better reflect this dependency, to avoid any potential misleading.

---

### Note · Authors · 2025-08-11

We thank all Reviewers and Area Chairs for their time and effort. We first present our key contributions, then summarize our reviewer-specific responses.

# Key Contributions

To narrow the gap between LoRA and full fine-tuning, LoRA-Pro explored aligning LoRA gradients with the full gradient. However, lacking a unique solution, they rely on an auxiliary variable and exhibit unstable performance. We address these challenges and propose a new PEFT method, AltLoRA, that achieves more effective gradient alignment with memory efficiency. Here, we discuss our key innovations:

  - Alternating Projection

We approximate the full gradient via alternating projections onto the low-rank subspaces $A$ and $B$. Also, we design a new momentum alignment mechanism that ensures momentum aligned within low-rank spaces under LoRA's memory constraints.

 - Theoretical Guarantees

Unlike LoRA-Pro, which lacks theoretical guarantees, we provide formal proofs that AltLoRA ensures stable feature learning and transformation invariance. We also establish a linear convergence rate independent of the condition number, allowing a wider range of learning rates.

 - Empirical Verification

Under the LoRA-Pro's setup, AltLoRA consistently outperforms existing LoRA-based methods and surpasses full fine-tuning on several benchmarks (MT-Bench, GSM8K).

# Reviewer-Specific Response

- Reviewer tXEi

We clarified that our methods remain stable when computing $(B^\top B + \lambda \mathbb{I})^{-1}$ during early training under standard strategies. In the context of gradient alignment, We provided theoretical justification for stable feature learning and transformation invariance, and conducted convergence analysis on an overparameterized two-layer ReLU model, following prior work. As in earlier LoRA studies, $\lambda$ is used for numerical stability but excluded from convergence analysis.


- Reviewer 4vyq

We added ablation results on momentum alignment showing clear benefits.

- Reviewer rKDu

We show that AltLoRA outperforms full fine-tuning with lower loss, faster convergence than most baselines, and higher accuracy in 5-epoch and LLaMA3-8B-Instruct experiments.

- Reviewer uqXj

We explain AltLoRA+ is built to accelerate convergence. Element-wise second-order momentum breaks invariance, and aligning it with the full gradient is possible but not consistently beneficial.



We believe our responses address all concerns and clearly convey the contributions and significance of our work.

---

### Decision · Program_Chairs · 2025-09-17

**Decision:**

Accept (poster)

**Comment:**

In this paper, the authors propose a new method for LoRA that aligns low-rank gradients with their full-rank counterparts, supported by theoretical analysis. The experimental results look promising. While reviewers initially raised concerns regarding the theoretical framework and empirical validation, the authors' rebuttal successfully addressed the majority of these issues. It is recommended that the authors incorporate the feedback from the rebuttal discussions into their revision.